# Mitochondrial support of persistent presynaptic vesicle mobilization with age-dependent synaptic growth after LTP

Heather L Smith[1], Jennifer N Bourne[2], Guan Cao[1], Michael A Chirillo[1], Linnaea E Ostroff[3], Deborah J Watson[1†], Kristen M Harris[1*]

[1]Department of Neuroscience, Center for Learning and Memory, Institute for Neuroscience, University of Texas at Austin, Austin, United States; [2]Department of Cell and Developmental Biology, University of Colorado Denver - Anschutz Medical Campus, Aurora, United States; [3]Center for Neural Science, New York University, Washington, New York

**Abstract** Mitochondria support synaptic transmission through production of ATP, sequestration of calcium, synthesis of glutamate, and other vital functions. Surprisingly, less than 50% of hippocampal CA1 presynaptic boutons contain mitochondria, raising the question of whether synapses without mitochondria can sustain changes in efficacy. To address this question, we analyzed synapses from postnatal day 15 (P15) and adult rat hippocampus that had undergone theta-burst stimulation to produce long-term potentiation (TBS-LTP) and compared them to control or no stimulation. At 30 and 120 min after TBS-LTP, vesicles were decreased only in presynaptic boutons that contained mitochondria at P15, and vesicle decrement was greatest in adult boutons containing mitochondria. Presynaptic mitochondrial cristae were widened, suggesting a sustained energy demand. Thus, mitochondrial proximity reflected enhanced vesicle mobilization well after potentiation reached asymptote, in parallel with the apparently silent addition of new dendritic spines at P15 or the silent enlargement of synapses in adults.

**\*For correspondence:** kmh2249@gmail.com

**Present address:** [†]QPS, LLC, Newark, United States

**Competing interests:** The authors declare that no competing interests exist.

## Introduction

Mitochondria are essential players in synaptic transmission. They produce more than 90% of all neuronal ATP (*Harris et al., 2012*), which is needed in many steps of the synaptic vesicle cycle (*Südhof, 1995*; *Chua et al., 2010*; *Sudhof and Rizo, 2011*). Mitochondria synthesize the excitatory neurotransmitter glutamate from glutamine (*Waagepetersoen et al., 2003*). The filling of synaptic vesicles with glutamate requires mitochondrial ATP (*Forgac, 2007*). Docking of synaptic vesicles does not appear to require ATP; however, priming them for subsequent release is an ATP-dependent process (*Yao and Bajjalieh, 2008*; *Verhage and Sørensen, 2008*). Following release, dissociation of the SNARE (soluble N-ethylmaleimide-sensitive factor attachment protein receptor) complex is an ATP-dependent process that reactivates monomeric SNARE proteins (*Sudhof and Rizo, 2011*). Endosome-based recycling of synaptic vesicles causes local depletion of ATP and increased production of ATP by mitochondria (*Rangaraju et al., 2014*). Mitochondria contain their own DNA; the proteome of presynaptic mitochondria is enriched in proteins that supply energy and are engaged in synaptic transmission, whereas the proteome of nonsynaptic axonal mitochondria clusters around metabolic and morphogenic proteins (*Völgyi et al., 2015*). Axonal mitochondria can move, although they typically remain stationary for periods of hours even after substantial activation (*Overly et al., 1996*; *Kang et al., 2008*; *Obashi and Okabe, 2013*). Interestingly, diffusion of ATP from mitochondria-containing boutons can sustain synaptic transmission at cultured hippocampal synapses without

mitochondria (*Pathak et al., 2015*). In addition, ATP from glycolysis can also support synaptic transmission under resting conditions, but mobilization of vesicles from the reserve pool requires mitochondrial ATP (*Verstreken et al., 2005*). Indeed, total vesicular release correlates well with the presence and volume of presynaptic mitochondria, suggesting their proximity may enhance synaptic transmission (*Sun et al., 2013*; *Ivannikov et al., 2013*).

Prior to weaning, at postnatal day 21 in rats, neurons have a low capacity for glucose utilization and instead use ketone bodies as their primary energy substrate (*Vannucci, 1994*; *Nehlig et al., 1988*; *Nehlig, 2004*). Unlike glucose metabolism, ketone body metabolism requires mitochondria (*Fukao et al., 2004*), suggesting there might be a stronger requirement for mitochondria to fuel synaptic vesicle cycling at younger ages. Changes in mitochondrial dimensions and the ultrastructure of matrix and cristae reflect altered energy demands (*Ishihara et al., 2009*; *Cagalinec et al., 2013*; *Hackenbrock, 1966*, *1968*; *Perkins et al., 2010*; *Packer, 1960*). Under resting conditions, mitochondria have loosely packed matrix and narrow cristae, whereas under conditions of high energy demand the matrix compacts and cristae widen (*Hackenbrock, 1966*, *1968*). These changes appear to reflect, for example, the action of mitochondrial dynamin-like proteins engaged in the metabolic processes (*Mannella, 2006*; *Patten et al., 2014*). Mitochondria also provide local sequestration of calcium that is important for enhanced vesicular release during post-tetanic potentiation lasting a few minutes (*Tang and Zucker, 1997*). Thus, the proximity and ultrastructure of presynaptic mitochondria should reliably reflect the capacity to mobilize vesicles and influence the duration of synaptic plasticity depending on the level of activation and the age of the animals.

Given their many critical functions, it was surprising to find that fewer than half of hippocampal presynaptic boutons contain a mitochondrion (*Shepherd and Harris, 1998*; *Kang et al., 2008*; *Pathak et al., 2015*; *Chavan et al., 2015*). This finding raised the question of whether synapses without presynaptic mitochondria can sustain robust and enduring changes in efficacy. To address this question, we analyzed hippocampal synapses that had undergone theta-burst stimulation (TBS) to produce long-term potentiation (LTP) in rat hippocampal area CA1 (*Bourne and Harris, 2011*; *Watson et al., 2016*). LTP is a cellular model of learning that produces enhancement of synaptic efficacy lasting hours to days or longer. Control synapses included those that had undergone test pulses in the same slice or from hippocampus that was unstimulated and perfusion-fixed in vivo. Comparisons were made between postnatal day 15, a pre-weaning age during peak synaptogenesis, and young adult rats at ages (55–70 days old) when intrinsic synaptogenesis has plateaued (*Harris et al., 1992*; *Fiala et al., 1998*; *Kirov et al., 2004*; *Semple et al., 2013*). Prior work has shown an LTP-related increase in synapse number at P15 (*Watson et al., 2016*), whereas, in the young adults, synapses enlarged at the expense of synaptogenesis by 120 min after the induction of LTP with TBS (*Bourne and Harris, 2011*; *Bell et al., 2014*). LTP is known to alter many presynaptic and postsynaptic mechanisms (*Bayazitov et al., 2007*; *Enoki et al., 2009*; *Huang et al., 2005*; *Ratnayaka et al., 2012*; *Zakharenko et al., 2001*; *Huganir and Nicoll, 2013*; *Arendt et al., 2015*; *Hill and Zito, 2013*). However, relationships between these changes in efficacy and the proximity of presynaptic mitochondria have not been investigated. Here, 3DEM was used to determine whether sustained synaptic plasticity was specific to mitochondria-containing presynaptic boutons.

## Results

At P15 and adult ages, control stimulation and theta-burst stimulation (TBS) produced independent physiological outcomes within the same slice (*Figure 1*). Hence, independent populations of synapses, having undergone TBS-LTP or unchanged responsivity under control stimulation, were compared within the same slice. The slices were prepared for 3DEM at key times with unbiased sampling of dendrites, axons, synapses, and mitochondria (Materials and methods, *Table 1*). Where appropriate, analyses were also completed for perfusion-fixed hippocampus as a basis for in vivo unstimulated values.

### Coordinated changes in the number of synapses and presynaptic boutons

At P15, control stimulation resulted in fewer dendritic spines, whereas LTP resulted in more spines than in perfusion-fixed hippocampus (*Figure 2A*, from *Watson et al., 2016*). To assess how the presynaptic boutons were affected, we used the unbiased brick analysis (*Fiala and Harris, 2001b*). All

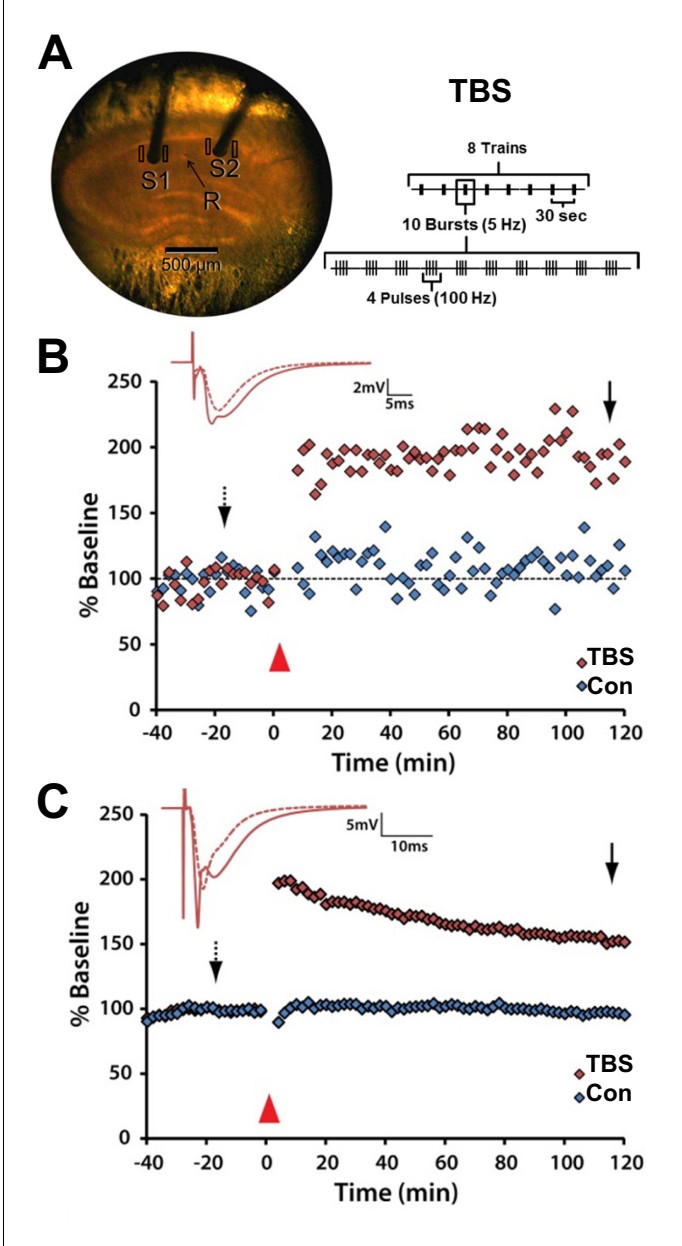

**Figure 1.** Stable responses during control stimulation and expression of LTP following TBS in *S. radiatum* of area CA1 in acute hippocampal slices. (**A**) P15 slice in interface chamber. Recording electrode (R) between two stimulating electrodes (S1 and S2) separated by more than 500 μm. Rectangles indicate the regions from which tissue was sampled for 3DEM. (**B**) Average responses after LTP was induced by TBS and average responses to control stimulation (Con) in P15 slices (n = 2 animals, modified from *Figure 1* of *Watson et al., 2016*). (Structural analyses were also obtained at 5 min and 30 min following TBS using experiments from *Watson et al., 2016*). (**C**) Average responses after LTP was induced by TBS and average responses to control stimulation (Con) from identically prepared slices at P55-70 (n = 2 animals; modified from *Figure 1* of *Bourne and Harris, 2011*). Red triangles indicate when TBS was applied. Arrows indicate times when the sample traces were obtained for the Con and TBS conditions.

boutons entering the inclusion volumes were followed through serial sections to determine whether they had a single synapse (*Figure 2B*), multiple synapses (*Figure 2C*), or no synapse (*Figure 2D*). At P15, single synapse boutons were significantly reduced under control stimulation relative to

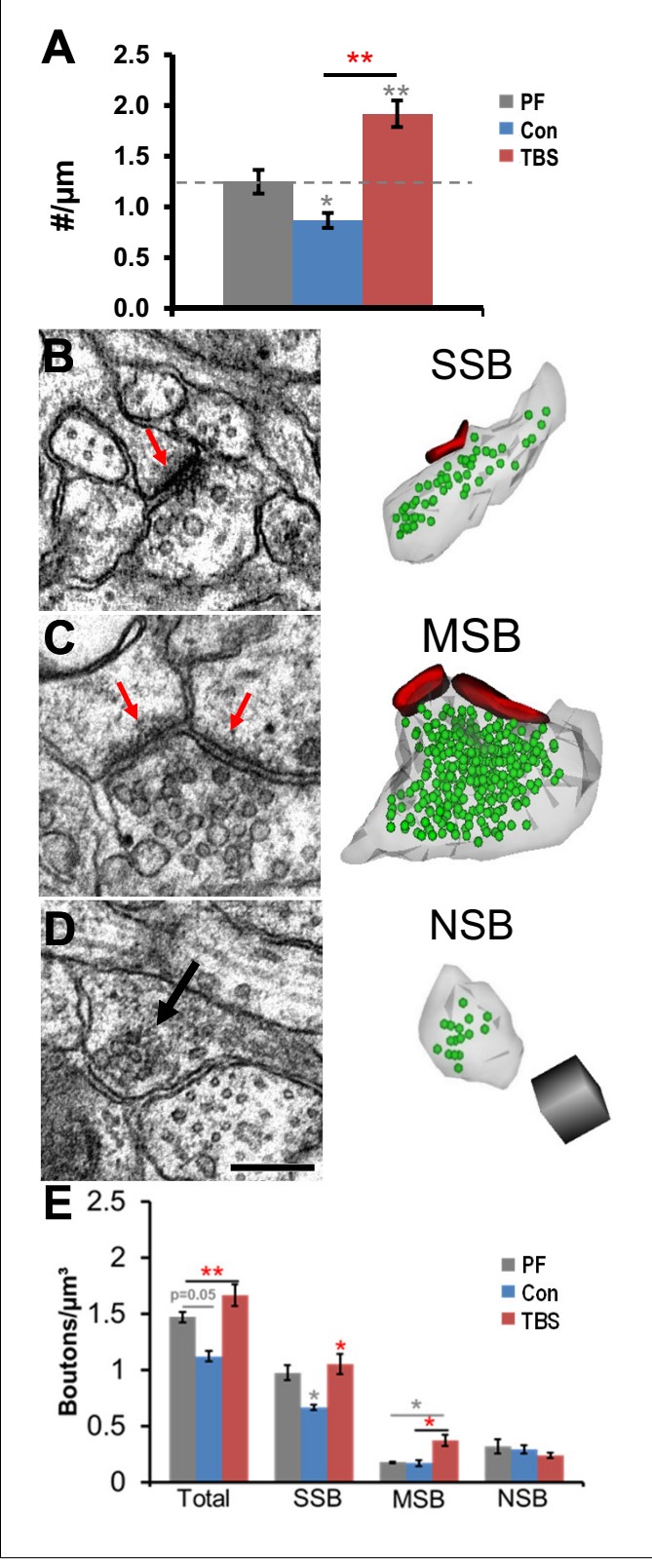

**Figure 2.** Maintenance of SSBs and increase in MSBs following induction of LTP by TBS at P15. (**A**) Spine density (#/µm length of reconstructed dendrite) from perfusion-fixed (PF) decreased under control stimulation (Con) and increased at 120 min after TBS induction of LTP ($F_{(2, 60)}$=26.45, p<0.0001, adapted from *Watson et al., 2016*). Example electron micrographs and 3D reconstructions of: (**B**) SSB with one PSD (red arrow), (**C**) MSB with two
*Figure 2 continued on next page*

*Figure 2 continued*

synapses (red arrows), and **D**) NSB (black arrow) with 16 vesicles but no postsynaptic partner. Reconstructions have PSDs (red), synaptic vesicles (green), and axonal plasma membranes (translucent gray). In **D** (for **B–D**), the scale bar is 0.25 μm and the scale cube is 0.25 μm on each side. (**E**) Overall, the frequency (boutons/μm$^3$ of unbiased bricks) of presynaptic boutons was significantly greater at 120 min following TBS induction of LTP, relative to PF and Con conditions (F = 11.312, df = 2, p<0.01). There were fewer SSBs under control conditions compared to PF, whereas SSB frequencies were maintained at PF levels following TBS and were significantly greater than controls (F$_{1,6}$ = 14.6, p=0.0087). MSB frequencies were greater following TBS than controls or PF (F$_{1,6}$ = 10.1, p=0.012). The NSB frequencies did not differ significantly across conditions (F$_{1,6}$ = 0.32, p=0.59). Two unbiased bricks (see Materials and methods) were analyzed per series for a total of 546 boutons across conditions (*Table 1*). Significant post hoc differences are indicated as *p<0.05, **p<0.01, ***p<0.001, and ****p<0.0001.

perfusion-fixed (*Figure 2E*). It is unlikely that these spines simply retracted, leaving intact presynaptic boutons, because the frequency of nonsynaptic boutons did not increase (*Figure 2E*). The frequency of single synaptic boutons in the LTP condition was comparable to the perfusion-fixed and greater than in the control condition, whereas multisynaptic boutons increased above control and perfusion-fixed conditions (*Figure 2E*). These findings suggest that at P15 the new spines formed after the induction of LTP made synapses primarily with pre-existing axonal boutons.

These results raised questions about age-dependent mechanisms of synaptic plasticity. In the rat hippocampus, P15 is an age when the rate of synaptogenesis is very high, while at P60 net synaptogenesis has plateaued in vivo (*Harris et al., 1992*; *Fiala et al., 1998*; *Kirov et al., 2004*; *Semple et al., 2013*). In adults, control stimulation produced spine recovery to perfusion-fixed levels, whereas TBS prevented spine formation and resulted in larger synapses by 120 min after the induction of LTP (*Bourne and Harris, 2011*; *Bell et al., 2014*). In adults, control stimulation resulted in more single synaptic boutons, whereas TBS induction of LTP prevented new single synaptic bouton formation and had no effect on multisynaptic boutons or nonsynaptic boutons (*Bourne et al., 2013*).

Here, we test the hypothesis that the availability of presynaptic mitochondria contribute to the lasting effects of LTP on presynaptic vesicle mobilization at both ages. We determine the in vivo content of mitochondria and synaptic vesicles in hippocampal presynaptic boutons at P15 and compare it to control and LTP conditions at 5, 30, and 120 min. Adult rat hippocampus has been previously characterized (*Shepherd and Harris, 1998*; *Bourne et al., 2013*), and these findings were reanalyzed in comparison to P15.

## Mitochondria-related differences in synapse size and vesicle composition in vivo

Axons reconstructed in perfusion-fixed hippocampus at P15 established the in vivo composition of mitochondria and vesicles as they relate to bouton configurations, and synapse size. Mitochondria were found both within synaptic boutons and in the neighboring nonsynaptic inter-bouton portions of the same axon (*Figure 3A,B*). About 75% of the presynaptic boutons were SSBs but only 25% of those contained mitochondria, whereas 25% were multisynaptic boutons and more than half (59%) of those contained mitochondria (*Figure 3C*). Comparable distributions occurred in adult hippocampus under control slice conditions, where 82% were single synaptic boutons and 36% contained mitochondria, while 18% were multisynaptic boutons with more than half (52%) containing mitochondria (from Table 2, *Shepherd and Harris, 1998*).

Individual PSD areas were larger when presynaptic boutons contained a mitochondrion, although this effect did not reach statistical significance for multisynaptic boutons (*Figure 3D*). At P15, both single synaptic boutons and multisynaptic boutons had more docked (*Figure 3E*) and nondocked (*Figure 3F*) vesicles when they contained a mitochondrion. These results are also consistent with findings from control slice conditions in adult hippocampus (*Shepherd and Harris, 1998*). Since these findings were the same for both types of presynaptic boutons, subsequent analyses do not distinguish between single synaptic and multisynaptic boutons.

These configurations might have resulted from mitochondria being preferentially located in larger axonal swellings. Then their greater association with more vesicles might have been due to chance

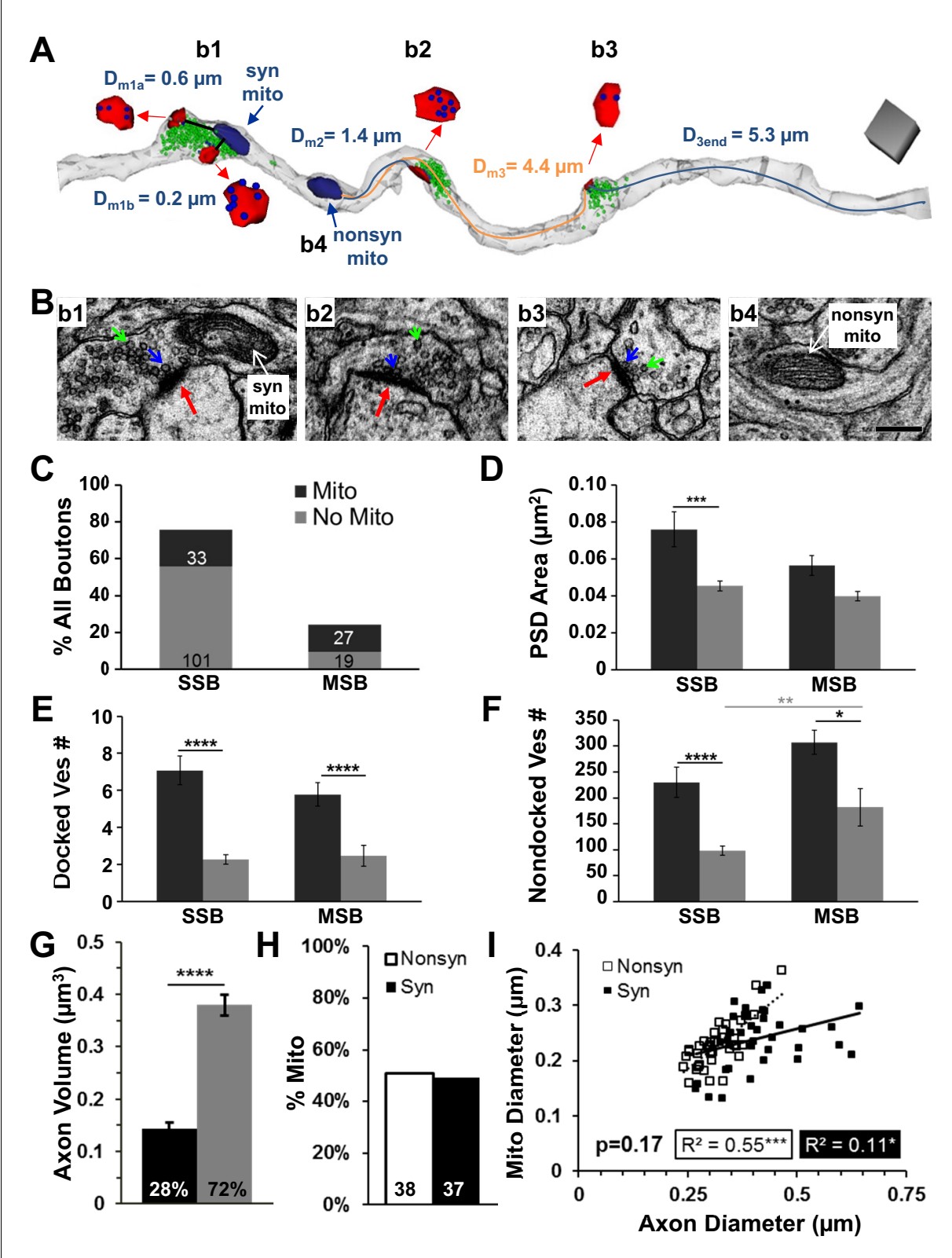

**Figure 3.** Presynaptic boutons with mitochondria had more vesicles than those without mitochondria in PF hippocampal area CA1 at P15. (**A**) Three neighboring presynaptic boutons located along the same axon (translucent gray). Bouton 1 (**b1**) was an MSB with two postsynaptic partners (PSDs, red) that contained a mitochondrion (navy), 303 nondocked vesicles (green), and 12 docked vesicles (blue, inset). Nearest distances from each PSD to the mitochondrion were less than 1 μm ($D_{m1a, 1b}$). For **b2**, the distance from the edge of the PSD to the nearest mitochondrion was 1.4 μm ($D_{m2}$, blue line),

*Figure 3 continued on next page*

*Figure 3 continued*

and it contained 124 nondocked vesicles and nine docked vesicles. The **b3** had 82 nondocked vesicles and two vesicles docked at the PSD which was 4.4 µm from the nearest mitochondrion (**b4**, nonsyn mito, $D_{m3}$, orange line) and 5.3 µm from the end of the axon reconstruction ($D_{3end}$). Scale cube is 0.5 µm on each side. (**B**) Representative EMs from four regions along an axon illustrating boutons with PSDs (red arrows), nondocked vesicles (green arrows), and docked vesicles (blue arrows) and a nonsynaptic inter-bouton region containing a mitochondrion (nonsyn mito). Scale bar is 0.25 µm in b4 for all four micrographs. (**C**) Of all the axonal boutons, 76% were SSBs and 24% were MSBs. Absolute numbers of boutons are in corresponding bars that also indicate the presence (black) or absence (gray) of a mitochondrion. (**D**) Overall, the PSD areas did not differ between synapses on MSBs and SSBs (n = 216, $F_{1,215}$ = 0.026, p=0.87). However, PSD areas were larger on SSBs when mitochondria were present (n = 140, $F_{1,138}$ = 15.4, p<0.001), but on MSBs this difference did not reach significance (n = 75, $F_{1,74}$ = 2.791, p=0.10). (**E**) The number of docked vesicles per PSD did not differ between MSBs and SSBs with or without mitochondria (n = 177, $F_{1,175}$ = 0.26, p=0.61); however, the number of docked vesicles per PSD was greater for both types of boutons with mitochondria than those without mitochondria (n = 177, $F_{1,175}$ = 65.1, p<0.001). (**F**) Boutons with mitochondria had more nondocked vesicles whether they were SSBs or MSBs (n = 180, $F_{1,178}$ = 16.2). Since the pool of nondocked vesicles could not be attributed to a specific PSD, the MSBs had more nondocked vesicles than SSBs, which reached significance for those boutons without mitochondria (gray line above the bars). (**G**) Axon segments containing mitochondria somewhere along their lengths were analyzed and the amount of axon volume with mitochondria was significantly less than the amount of axon volume without mitochondria (n = 81, ANOVA, $F_{1,161}$ = 104.1, p<0.0001). (**H**) Of all axonal mitochondria in the perfusion fixed condition from the brick analyses (see *Figure 2E*), 51% of mitochondria resided in nonsynaptic regions (Nonsyn), while 49% resided in synaptic boutons (Syn). (**I**) Axonal and mitochondrial diameters were correlated and this relationship did not differ significantly between mitochondria located in synaptic or nonsynaptic regions of the axons from (**H**) (n = 75, ANCOVA, $F_{1,72}$ = 1.91, p=0.17). Significant post hoc differences are indicated as *p<0.05, **p<0.01, ***p<0.001, and ****p<0.0001.

because the larger synaptic boutons contained more vesicles. We performed several analyses to address this concern. First, we computed that the portion of axonal volume occupied by mitochondria (28%) was markedly less than the portion of axonal volume without mitochondria (72%, *Figure 3G*). Next, we showed that mitochondria were equally distributed between nonsynaptic inter-bouton regions (51%) and synaptic boutons (49%, *Figure 3H*). As predicted, we found that axonal diameters were significantly wider at synaptic boutons (0.41 ± 0.015 µm) than at nonsynaptic regions (0.32 ± 0.008 µm, n = 75, $F_{1,71}$ = 37.5, p<0.001). However, mitochondrial diameters were similar at synaptic (0.24 ± 0.009 µm) and nonsynaptic (0.23 ± 0.007 µm) regions (n = 75, $F_{1,71}$ = 1.25, p=0.27). Finally, the mitochondria fit into all regions of the axon (*Figure 3I*). These results suggest that mitochondria were not restricted from entering even the narrow inter-bouton regions of the axon, as would be required since they can move along axons. The findings also show that mitochondria were not uniformly distributed throughout the axon, as the vast majority of the axon length and volume contained no mitochondria. Furthermore, as shown in subsequent analyses, synapses spanning the full range of sizes (indicated by the presynaptic vesicle counts and PSD areas in *Figures 4*, *7* and *8*) could be associated with presynaptic boutons that contained mitochondria as well as with those that did not contain mitochondria. Thus, the significant effects of mitochondrial proximity on changes in synapse size and presynaptic composition reported below do not appear merely to be due to a chance restriction of mitochondria to larger synaptic boutons.

## Influence of distance from mitochondria on synapse size and vesicle content

The diffusion of ATP in cultured axons is somewhat limited by rapid ATP utilization (*Rangaraju et al., 2014*; *Sun et al., 2013*); nevertheless, a diffusion coefficient of 530 µm²/s has been measured (*Nicolay et al., 2001*). Under these conditions, ATP could diffuse from a stationary mitochondrion to a synapse located up to three microns away in about nine milliseconds. As a proxy for ATP availability, we measured distances from the center of each PSD to the nearest edge of the nearest mitochondrion ($D_m$, *Figure 3A*). These analyses were done first in perfusion-fixed hippocampus at P15 and applied subsequently to all conditions and ages.

$D_m$ was plotted relative to the number of nondocked or docked vesicles and total PSD area per bouton (*Figure 4A–C*). The wedge-shaped distributions suggested there might be a critical threshold distance at which the relationship between synapse efficacy and size versus mitochondrial proximity abruptly changes. A statistical threshold is defined as a point at which small changes in an environmental driver induce large changes in a system (*Peter et al., 2006*). Piecewise regression functions were fitted to each scatterplot to determine whether there is a threshold distance beyond

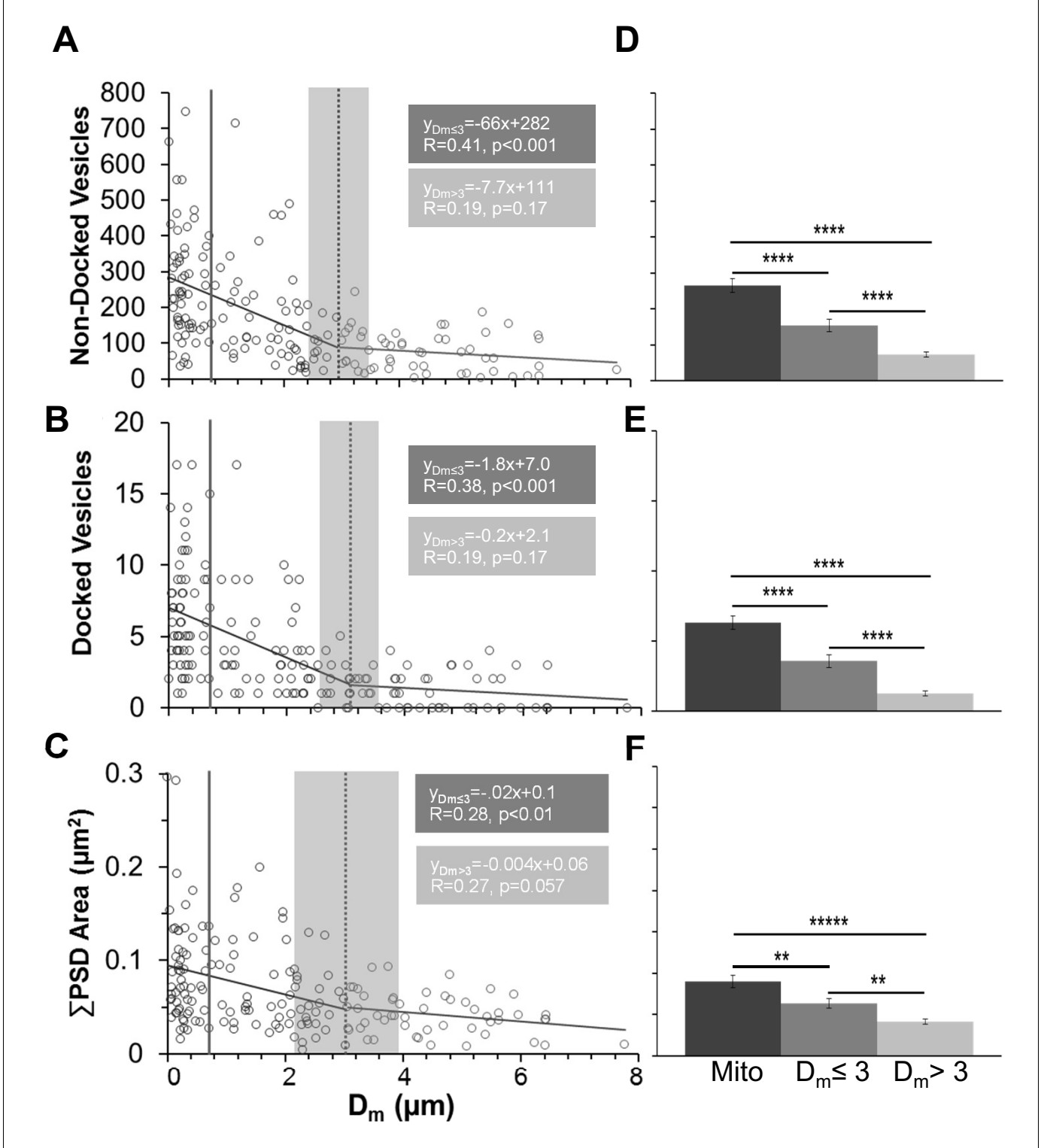

**Figure 4.** Synapses had fewer vesicles and PSD areas were smaller as distance from an axonal mitochondrion increased. (A) Nondocked vesicles versus mitochondrial distance ($D_m$) had a statistically significant breakpoint at $2.97 \pm 0.67$ μm ($t(165) = 4.4$, $p<0.001$). (B) The breakpoint for docked vesicles per PSD area versus mitochondrial distance was $3.06 \pm 0.62$ μm ($t(161) = 4.92$, $p<0.001$). (C) The breakpoint for PSD area per bouton ($\Sigma$PSD across MSBs) vs. $D_m$ was $3.02 \pm 1.24$ μm ($t(165) = 2.44$, $p=0.015$). The mitochondria had lengths and hence were considered to be inside the bouton when any portion of the mitochondrion was within the vesicular domain and distances from the center of the PSD were shorter than the solid gray line positioned at 0.91 μm (in A–C), which was the longest measured distance from the edge of a PSD to a mitochondrion located within the vesicle cloud of a presynaptic

*Figure 4 continued on next page*

*Figure 4 continued*

bouton. Breakpoints in **A–C** are shown as mean (dashed line) ± s.e.m. (gray shading) from the regression analysis. (**D**) The mean number of nondocked vesicles was greater in boutons with a mitochondrion (mito) than at $D_m \leq 3$ μm, which was greater than $D_m > 3$ μm (n = 177, $F_{2,175}$ = 46.3, p<0.001). (**E**) The mean numbers of docked vesicles per PSD area was greater in boutons with a mitochondrion than at $D_m \leq 3$ μm, which was greater than $D_m > 3$ μm (n = 183, $F_{2,181}$ = 55.0, p<0.001). (**F**) The mean $\sum$PSD area was also greater when presynaptic boutons had a mitochondrion than when $D_m \leq 3$ μm, which was greater than $D_m > 3$ μm (n = 175, $F_{2,175}$ = 19.6, p<0.001). All these data are from perfusion-fixed hippocampus at P15. y-Axis labels in **A–C** are also for **D–F**, and similarly the x-axis labels at the bottom apply to all graphs above them. Significant post hoc differences are indicated as *p<0.05, **p<0.01, ***p<0.001, and ****p<0.0001.

which synaptic efficacy, as related to presynaptic vesicle number or synapse area, is no longer correlated with mitochondrial proximity.

For nondocked and docked vesicles as well as PSD areas, the distance threshold was ~3 μm (*Figure 4A–C*). Only correlations at shorter distances from the breakpoint were statistically significant (*Figure 4A–C*). Thus, boutons could be separated into three categories: those that contained a mitochondrion (mito), those that had the nearest mitochondrion located outside the bouton but within 3 μm of the synapse center ($D_m \leq 3$), and those that had the nearest mitochondrion located at distances greater than 3 μm from the synapse ($D_m > 3$).

At P15, approximately a third of the boutons from perfusion-fixed hippocampus had synapses in each categorical range, with 33% containing mitochondria, 31% having $D_m \leq 3$, and 36% having $D_m > 3$. Boutons containing a mitochondrion had more nondocked vesicles (*Figure 4D*), docked vesicles (*Figure 4E*), and larger total PSD areas (*Figure 4F*) than those with $D_m \leq 3$ μm, which were all greater than those with $D_m > 3$ μm. These results suggest that the proximity of a mitochondrion has a strongly positive influence on synaptic size and vesicle number.

## Stability in position of presynaptic mitochondria across conditions at P15

Mitochondria tend to be stationary at synapses for hours to days (*Obashi and Okabe, 2013*). ATP can remain within the mitochondria-containing boutons (*Sun et al., 2013*) or diffuse from mitochondria-containing to non-mitochondria-containing boutons (*Palthak et al., 2015*) of cultured hippocampal axons. We hypothesized that boutons with mitochondria would have more resources available to undergo and sustain changes in synaptic efficacy. To test this hypothesis, axons were reconstructed from each of the time points for control and LTP conditions (*Figure 5*). About 25–42% of the synapses had presynaptic mitochondria located within the three categories of proximity (*Figure 5*). These proportions were unchanged relative to perfusion-fixed levels under control conditions ($\chi^2(6)$ = 5.35, p=0.5) and did not differ significantly at any time after TBS. By 120 min in the TBS condition, the Dm > 3 was greatest, up from 26% to 40%, suggesting that the new spine synapses were formed at longer distances from mitochondria, an issue that will be discussed again below (see *Figure 9*).

## Decrease in presynaptic vesicles of mitochondria-containing boutons after TBS at P15

Next, we tested whether the presence of a mitochondrion influenced changes in presynaptic vesicles during control stimulation or following TBS at P15. Neither docked nor nondocked vesicle numbers differed significantly between control stimulation and perfusion-fixed conditions at any time point or mitochondrial proximity (*Figure 6*, compare blue to gray bars). Like the perfusion-fixed condition, presynaptic boutons with mitochondria contained significantly more docked and nondocked vesicles than boutons without mitochondria at all three times after control stimulation (*Figure 6*, blue stars and lines). Thus, although control stimulation eliminated spines and boutons by 120 min (See *Figure 2* above), the distribution and number of vesicles remained highest in boutons that contained a mitochondrion and was unaffected by control stimulation.

At P15, there were time-dependent effects of TBS on presynaptic vesicles that also reflected mitochondria proximity. At 5 min (*Figure 6A*), there were no significant effects of TBS on docked or nondocked vesicle numbers regardless of mitochondrial proximity (*Figure 6A1 and A2*). At 30 min (*Figure 6B*), the boutons with mitochondria had fewer docked vesicles per synapse than in time-

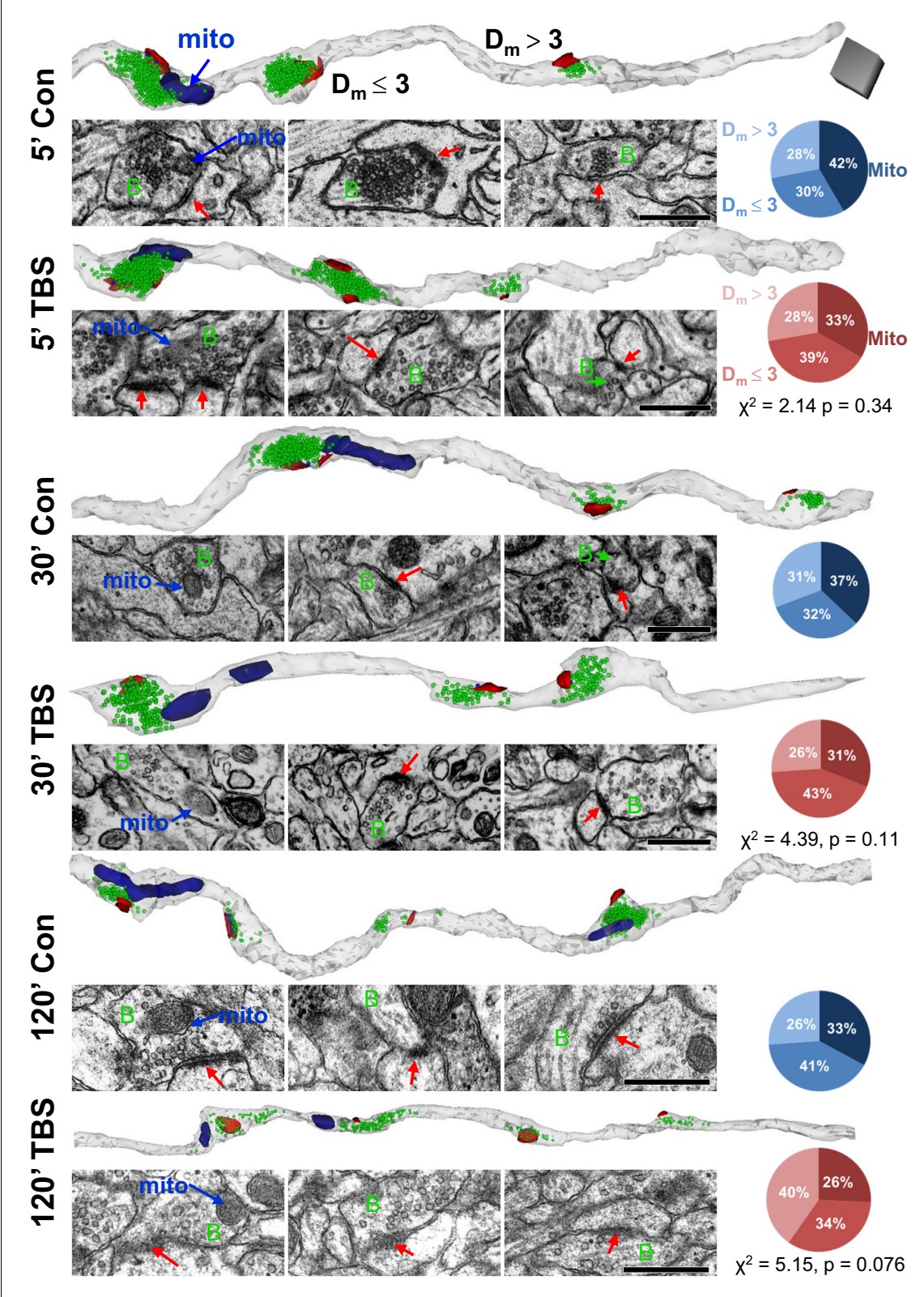

**Figure 5.** The relative distribution of mitochondrial distances from synapses was unchanged for control and LTP conditions at P15. Representative axons are shown from the control (Con) and TBS sites for each time point (5', 30', 120' = minutes) after the induction of LTP at P15. Each axon is oriented to illustrate sequentially a presynaptic bouton with a mitochondrion (mito), boutons with PSDs located within 3 μm of a mitochondrion ($D_m \leq 3$) or beyond ($D_m > 3$). For each time and condition, a 3D reconstruction of the axons (gray) illustrates PSDs (red), nondocked vesicles (green), and

*Figure 5 continued on next page*

Figure 5 continued

mitochondria (navy). Sample EMs are positioned beneath each bouton to illustrate their PSDs (red arrows in postsynaptic side), presynaptic boutons (green B), and in the leftmost column the mitochondria (blue arrows). Pie charts indicate the proportions of boutons that reside in each category (Mito, $D_m \pounds$ 3, $D_m > 3$) for the corresponding condition and time point. Scale cube (dark gray) in top row is 0.5 µm on each side for all 3D reconstructions, and scale bar is 0.5 µm for all EMs.

matched control or perfusion-fixed conditions (*Figure 6B1*), but the nondocked reserve pool was unchanged (*Figure 6B2*). At 120 min (*Figure 6C*), there were fewer docked vesicles at synapses with presynaptic mitochondria or $D_m \leq 3$ µm in the TBS relative to the control or perfusion-fixed conditions (*Figure 6C1*). In contrast with earlier time points, the reserve pool of nondocked vesicles was also diminished at 120 min in the TBS relative to the control or perfusion-fixed conditions when the presynaptic boutons contained a mitochondrion (*Figure 6C2*).

As indicated in *Figure 2* above, small synapses were added in the LTP condition at P15, and here we wanted to know whether vesicles proliferated or redistributed to the added synapses. The correlation between vesicle numbers and synapse size was strong for all conditions, whether or not a mitochondrion was present in the presynaptic bouton (*Figure 7*). When a mitochondrion was present in the presynaptic bouton, the number of docked and nondocked vesicles was significantly decreased across synapses of all sizes at both 30 min (*Figure 7A,B*) and 120 min (*Figure 7E,F*) following the induction of LTP relative to control stimulation. When no mitochondrion was present in the presynaptic bouton, neither docked nor nondocked vesicle numbers differed significantly between TBS and control conditions at 30 min (*Figure 7C,D*) or 120 min (*Figure 7G,H*). Synapses without mitochondria were more likely to have no presynaptically docked vesicles at both time points under control conditions (7C, G). More of the synapses without mitochondria lacked docked vesicles at 120 min after induction of LTP with TBS than under control conditions (*Figure 7G*). This increase in presynaptic boutons lacking docked vesicles is coincident with the formation of new small dendritic spines (*Watson et al., 2016*).

Together these findings support the hypothesis that the induction of LTP at P15 involved a sustained increase in mobilization of the reserve pool of presynaptic vesicles in the subset of boutons containing a mitochondrion. While this reduction was present at all synapse sizes, it was most pronounced at larger synapses, suggesting that the newly formed small synapses had less vesicular mobilization. Furthermore, the increase in boutons without any docked vesicles suggests that the newly formed synapses had not yet built presynaptic docking sites.

## Presynaptic mitochondria also sustain vesicle mobilization following TBS in adults

Neurons from post-weanling and adult rats can produce ATP via glycolysis, a process that does not require mitochondria. Thus, mitochondria might not be as critical for elevated release or vesicle mobilization following the induction of LTP in adults versus P15, a pre-weaning age. To test this hypothesis, presynaptic vesicles were compared across boutons with and without mitochondria in adult hippocampal slices that had undergone the same TBS or control stimulation protocols described above for P15 (*Bourne et al., 2013*). Since the greatest effects were at 120 min, the rest of the analyses are from that timepoint.

At 120 min, the total number of boutons per cubic micron was about 50% less in the TBS relative to control conditions (*Figure 8C*, *Bourne et al., 2013*). Additionally, more of the presynaptic boutons contained mitochondria in adults than at P15 for both control ($\chi^2$ = 14.3, df = 1, p<0.001) and TBS conditions ($\chi^2$ = 33.7, df = 1, p<0.00001; compare *Figure 8D,E* with *Figure 5* at 120 min). In the adults, like P15, presynaptic boutons with a mitochondrion had more vesicles than those without a mitochondrion (*Figure 8F1, G1*; $F_{2,226}$ = 47.2, p<0.001). Like P15, the PSD areas of the adult synapses were larger when their presynaptic boutons contained a mitochondrion (0.074 ± 0.0058 µm$^2$) than when they did not contain a mitochondrion (0.040 ± 0.0025 µm$^2$; $F_{2,226}$ = 26.3, p<0.0001). Unlike P15, however, adult synapses rarely lacked docked vesicles whether or not a mitochondrion was present (*Figure 8F2, G2*). Also in the adults, the docked and nondocked vesicle numbers were well correlated with PSD area, whether or not the bouton contained a mitochondrion (*Figure 8F1-3, G1-3*). Furthermore, there was a marked decrease in vesicle numbers of mitochondria-containing

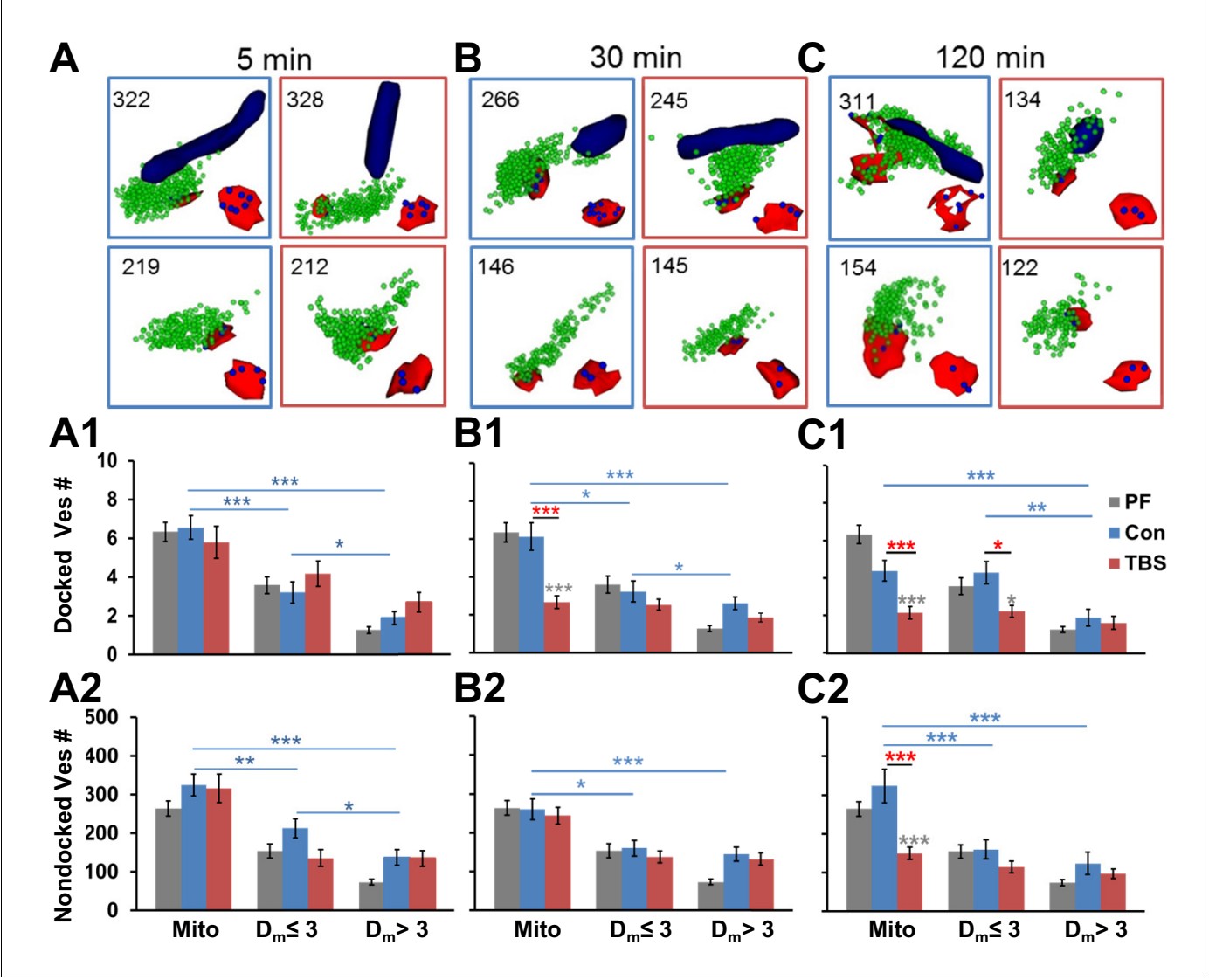

**Figure 6.** Mobilization of presynaptic vesicles was restricted to presynaptic boutons with or near mitochondria. Representative 3D reconstructions of boutons with mitochondria (navy, top row) and without mitochondria (second row) with PSDs (red), docked vesicles (blue), and nondocked vesicles (green) at (A) 5 min, (B) 30 min, and (C) 120 min post-TBS. Light blue boxes enclose the control conditions, and red boxes enclose the TBS conditions at each time point. Total vesicle number is at top left corner, and PSDs with docked vesicles are reconstructed in bottom right corner of each box. At 5 min, Con conditions had (A1) more docked vesicles ($F_{2,94}$ = 20.4, p<0.001) and more (A2) nondocked vesicles ($F_{2,126}$ = 14.7, p<0.01) in presynaptic boutons with mitochondria (Mito) or at $D_m \leq 3$ μm (blue bars and stars), but there were no LTP effects. At 30 min, (B1) docked vesicles decreased post-TBS, relative to Con and PF conditions only when presynaptic boutons contained a mitochondrion ($F_{1,138}$ = 13.7, p<0.001, red stars). In addition, Con boutons with Mito or $D_m \leq 3$ had more docked vesicles ($F_{2,91}$ = 10.9, p<0.001) and nondocked vesicles ($F_{2,132}$ = 10.1, p<0.001, blue stars and bars). At 120 min, (C1) docked vesicles were reduced in boutons with mitochondria ($F_{1,106}$ = 13.9, p<0.001) and when $D_m$ £ 3 ($F_{1,102}$ = 9.06, p<0.01) relative to control and PF conditions. At 120 min when $D_m \leq 3$, boutons have more docked vesicles in Cons ($F_{2,63}$ = 4.27, p=0.018). Nondocked vesicles (C2) were reduced in boutons with mitochondria relative to Con and PF levels ($F_{1,122}$ = 20.2, p<0. 01). Furthermore, Con boutons contain more nondocked vesicles when mitochondria are present ($F_{2,81}$ = 6.93, p<0.01). Significant post hoc differences are indicated as *p<0.05, **p<0.01, ***p<0.001, and ****p<0.0001.

boutons across synapses of all sizes at 120 min after TBS relative to control stimulation in adults (*Figures 8F2* and *3*). However, there was also a small decrease in vesicle number in boutons without a presynaptic mitochondrion in adults (*Figures 8G2* and *3*). Thus, enhanced vesicular mobilization was greatest in mitochondria-containing presynaptic boutons at both ages but also occurred in

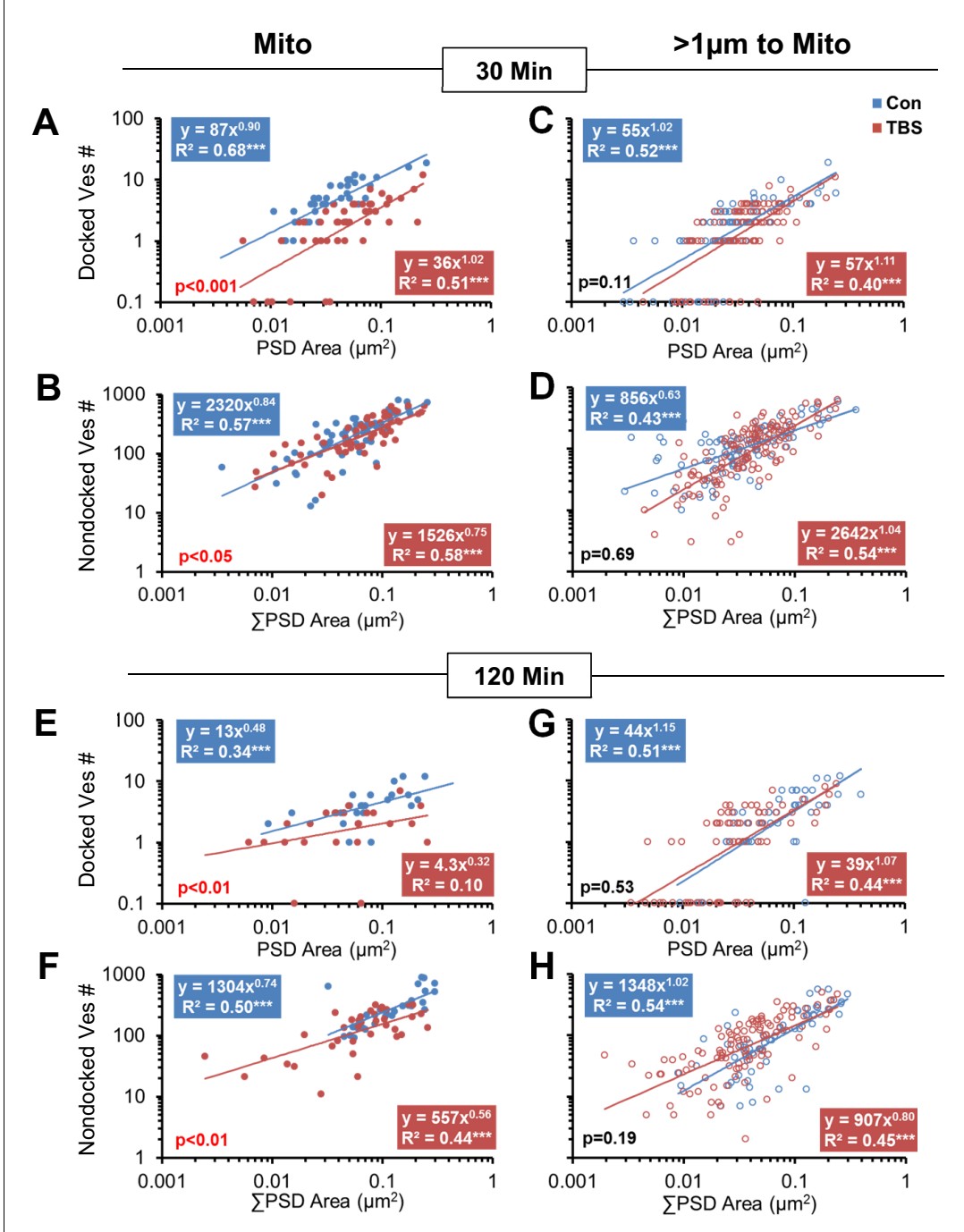

**Figure 7.** Effect of synapse size and mitochondrial presence on vesicle mobilization following TBS at P15. Synapses with mitochondria located among the presynaptic vesicles (Mito) were compared to those without mitochondria (>1 μm to Mito). At 30 min following TBS, boutons with mitochondria had (A) fewer docked vesicles ($F_{1,81} = 61.4$, $p<0.001$) and (B) fewer nondocked vesicles ($F_{1,108} = 5.03$, $p=0.021$) relative to control conditions. In contrast, boutons with no mitochondria showed no significant differences in (C) docked ($F_{1,169} = 2.64$, $p=0.11$) or (D) nondocked vesicles ($F_{1,227} = 0.17$, $p=0.69$) between TBS and control conditions at 30 min. Furthermore, under control conditions, (A) fewer boutons with mitochondria had zero docked vesicles than (B) boutons with no mitochondria (F; $F_{1,107} = 3.87$, $p<0.05$), but this mitochondria-related difference ($F_{1,145} = 0.57$, $p=0.45$) did not hold in the TBS condition. At 120 min following TBS, boutons with mitochondria had (E) fewer docked ($F_{1,44} = 12.4$, $p=0.001008$) and (F) fewer nondocked vesicles ($F_{1,63} = 7.84$, $p=0.007$) relative to control conditions across the distribution of synapse sizes. In contrast, boutons with no mitochondria showed no significant differences between the TBS and control conditions in the number of (G) docked vesicles ($F_{1,116} = 0.41$, $p=0.53$) or (H) nondocked vesicles ($F_{1,169} = 1.71$, $p=0.19$). Furthermore, fewer boutons (E) with mitochondria lacked docked vesicles than those (F) without mitochondria ($F_{1,92} = 4.2$, $p<0.05$) at 120 min following TBS, whereas this difference did not reach significance under the control conditions ($F_{1,72} = 3.01$, $p=0.08$). In addition, (G)

*Figure 7 continued on next page*

*Figure 7 continued*

more of the boutons without mitochondria lacked docked vesicles at 120 min after TBS than under control conditions ($F_{1,118}$ = 4.9, p<0.05). The x-axis represents individual PSD areas for docked vesicles (**A**, **C**, **E**, **G**) and summed PSD area for nondocked vesicles to account for MSBs (**B**, **D**, **F**, **H**). The results of the ANCOVAs are indicated in the lower left corner of each graph, whereas the results from ANOVAs concerning the differences among conditions for boutons lacking docked vesicles are summarized in this figure legend.

those without mitochondria in adult hippocampus, contrasting with P15 where this mobilization only occurred in the mitochondria-containing presynaptic boutons.

## Changes in mitochondrial dimensions following induction of LTP by TBS

Increases in synaptic vesicle mobilization have been associated with increased demand for ATP (*Rangaraju et al., 2014*). Mitochondria undergo fission to accommodate fluctuating energy demands. Then as mitochondrial resources are depleted they fuse with new mitochondria delivered from the soma, resulting in their elongation (*Cagalinec et al., 2013*; *Chang et al., 2006*; *Huang et al., 2015*; *Packer, 1960*; *Sun et al., 2013*). To assess whether the sustained vesicular mobilization resulted in mitochondrial fission or fusion, we measured mitochondria in axons from the TBS and control conditions at 120 min for both ages (*Figure 9A,B*). Mitochondrial volume was reduced at P15 and increased in adults in the TBS versus control conditions (*Figure 9C*). Mitochondrial diameters did not differ significantly between the TBS and control conditions for either age, although they were slightly wider in the adults than at P15 (*Figure 9D*). Mitochondrial lengths were shorter in the control than in the TBS condition at P15 but were longer in the control condition in adults (*Figure 9E*). The frequency of mitochondria was greater in adults than at P15 under control conditions, but was much less in the adults after TBS (*Figure 9F*).

Together these findings suggest that by 120 min after TBS-LTP, mitochondria became smaller without fission at P15 and become larger through fusion in adults. This differential fission/fusion response suggests that at P15 the mitochondrial resources were depleted before new mitochondria could be generated or transported to the active sites to replenish those resources. In contrast, the findings from adults suggest that mitochondria fusion occured when the frequency of presynaptic boutons decreases following TBS. Furthermore, in adults, but not at P15, the elevated vesicular mobilization also occurred in boutons without mitochondria, which might have been feasible because glycolysis does not require mitochondria and can be a source of ATP in adults but not at P15 (*Vannucci, 1994*).

These data also show that at P15 the axonal mitochondria occurred on average at distances of 1 per 7.13 ± 0.48 microns after control stimulation, which was not significantly different from 1 per 8.07 ± 0.34 microns following TBS (*Figure 9F*). Thus, the new dendritic spine synapses were likely to have occurred with presynaptic boutons that did not contain a mitochondrion because the distribution of mitochondria did not change significantly (Compare *Figure 5*, above).

In adults, mitochondria occurred on average at distances of 1 per 3.6 ± 0.25 microns following control stimulation, but only 1 per 6.10 ± 0.40 microns following TBS (*Figure 9F*). Examination of the plots in *Figure 8* above suggests enlargement of adult synapses occurred at boutons with and without mitochondria. Similarly, presynaptic boutons sustained during control stimulation also occurred with and without mitochondria. In addition, as both synapses and mitochondria were reduced in frequency, the proportion of synapses that contained mitochondria was unchanged, suggesting that mitochondrial demand could partially be determined by the number of synapses that require their support.

## LTP-Associated changes in mitochondrial structure reflect enhanced ATP production

Changes in the internal structure of mitochondria in response to increased demand for energy include swelling of cristae and reduced density of the matrix (*Packer, 1960*; *Hackenbrock, 1966*; *Perkins and Ellisman, 2011*). The mitochondrial configurations described in these papers and measures of cristae widths were used as proxies to investigate whether changes in energy demand might have occurred at 120 min after TBS. Under normal resting conditions, the tricarboxylic acid cycle substrates are ample, energy demand is low, and mitochondria assume an orthodox configuration

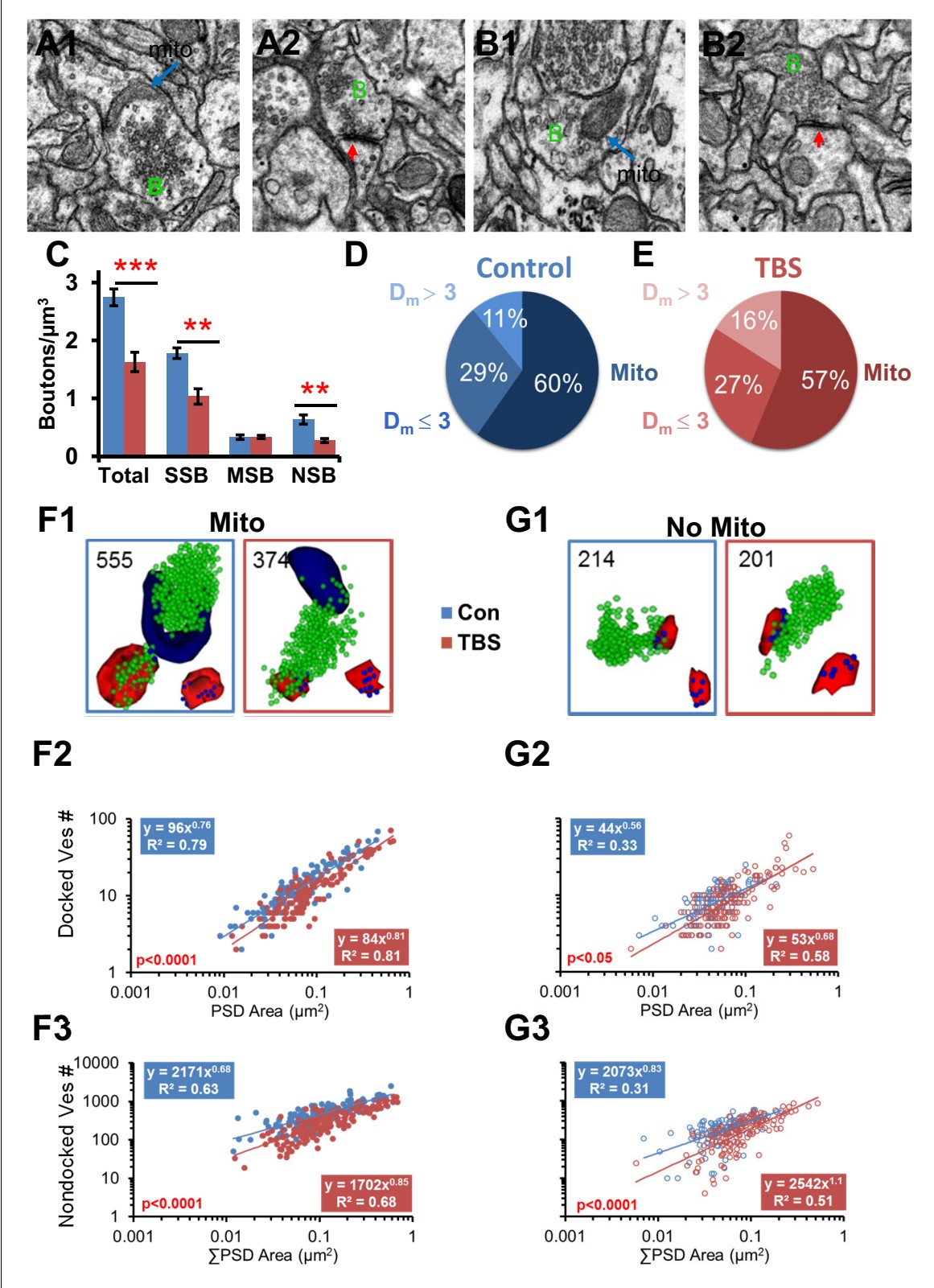

**Figure 8.** Vesicle mobilization was also greatest in presynaptic boutons that contained a mitochondrion at 120 min following TBS in adult hippocampus. (A–B) Representative EMs of presynaptic boutons (green B) from 120 min control (A1 mito, A2 no mito) and TBS (B1 mito, B2 no mito) conditions (red arrows, PSDs in A1, B2, scale bar is 0.5 μm). (C) Overall, the total number of boutons per μm$^3$ was decreased under TBS conditions relative to control (n = 6 vol bricks per condition, see *Table 1*, $F_{1,8}$ = 30.8, p<0.001). There were fewer single synaptic boutons (SSB, n = 6, $F_{1,8}$ = 22.6, *Figure 8 continued on next page*

*Figure 8 continued*

p<0.01) and nonsynaptic boutons (NSB, n = 6, $F_{1,8}$ = 13.9, p<0.01) 2 hr after the onset of TBS, but the multisynaptic boutons were unchanged (MSB, n = 6, $F_{1,8}$ = 0.22, p=0.65). Proportions of boutons that contained mitochondria (Mito) or were $D_m \leq 3$ or $D_m > 3$ from a mitochondrion under control (D) and TBS (E) conditions were unchanged in the LTP relative to the control condition (n = 518, $\chi^2$=2.98, p=0.23). Representative reconstructions of synapses from adult hippocampal area CA1 at the 120 min (PSDs, red, nondocked vesicles, green, and docked vesicles, blue, lower right corner) for boutons (F1) with mitochondria (navy, Mito) and (G1) without mitochondria (No Mito). (F2) Docked vesicles decreased at 120 min after TBS relative to Con in boutons with mitochondria ($F_{1,244}$ = 27.2, p<0.001), and (G2) to a lesser degree in boutons without mitochondria ($F_{1,207}$ = 4.72, p<0.05) across synapses of all sizes. (F3) Nondocked vesicles also had a pronounced decrease in the TBS condition relative to Con in boutons with mitochondria ($F_{1,297}$ = 112.1, p<0.001) and (G3) to a lesser degree in boutons without mitochondria ($F_{1,250}$ = 35.8, p<0.001).

that is characterized by homogeneously narrow cristae and uniformly distributed matrix (*Figure 10A1*). When the concentration of ADP increases with elevated local energy demand, the mitochondria assume non-orthodox configurations, with wider cristae and more compact or irregular matrices (*Figure 10A2–4*). Swollen mitochondria (*Figure 10A5*) occur under hypoxic conditions and were rare under all conditions reported here.

Since synaptic and nonsynaptic mitochondria have different proteomes (*Völgyi et al., 2015*), we wanted to know whether the ultrastructure might reveal differences in energy utilization between synaptic and nonsynaptic mitochondria. At P15, less than 50% of all mitochondria were synaptic (i.e. located within the presynaptic bouton), whereas in the adults more than 80% of all mitochondria were synaptic (*Figure 10B*). Under control stimulation conditions, more than 70% of axonal mitochondria were in the orthodox configuration in both P15 and adult axons (*Figure 10C*). The frequency of mitochondria in the orthodox configuration decreased by about half at 120 min post-TBS in both P15 and adult slices relative to control stimulation and were replaced by dramatic increases in mitochondria with the condensed, intermediate, and mixed configurations (*Figure 10D*). The LTP-related shifts in mitochondrial configurations resulted in wider cristae at both synaptic and nonsynaptic mitochondria for the TBS vs. control conditions at both ages (*Figure 10E*). These changes in mitochondria structure are consistent with an elevated demand for presynaptic ATP at 120 min after TBS-LTP.

## Discussion

The evoked response to control stimulation and level of potentiation remained stable from 5 to 120 min after TBS at both ages, raising the question of whether the dramatic changes observed in synapse number and structure would be functionally silent. At P15, control stimulation eliminated dendritic spines, whereas following TBS there was a nearly 50% increase in spines, which served to sustain single synaptic boutons and increase multisynaptic boutons. In contrast, in adults the spinogenesis induced by control stimulation was blocked by the TBS-related synapse enlargement (*Bourne and Harris, 2011*; *Bell et al., 2014*). At P15, less than 30% of the presynaptic boutons contained mitochondria at 2 hr following TBS, suggesting that TBS-related spinogenesis mostly involved presynaptic boutons without mitochondria. Since only the mitochondria-containing boutons had elevated vesicle mobilization at P15, the TBS-related spinogenesis likely occurred at boutons lacking enhanced vesicle mobilization. This arrangement could account for the stable level of potentiation at 2 hr despite the addition of new spines. In adults, more boutons contained mitochondria under both control and TBS conditions than at P15, and TBS resulted in fewer small spines and SSBs. The number of mitochondria also decreased in adult axons following TBS. These results suggest that in adults, but not at P15, mitochondrial and synaptic frequencies are coupled.

In adults, the TBS-related synapse enlargement and vesicle mobilization occurred at boutons with or without mitochondria. However, much of the growth in the PSD involved the addition of nascent zones that had PSDs but lacked presynaptic vesicles, and hence were likely to be unresponsive to vesicular release (*Bell et al., 2014*). Thus, at both ages, synaptogenesis, synaptic growth, and vesicular mobilization either following control stimulation or TBS appear to have involved functionally silent processes. What mechanisms could be engaged and what are the functional consequences of building these silent synapses?

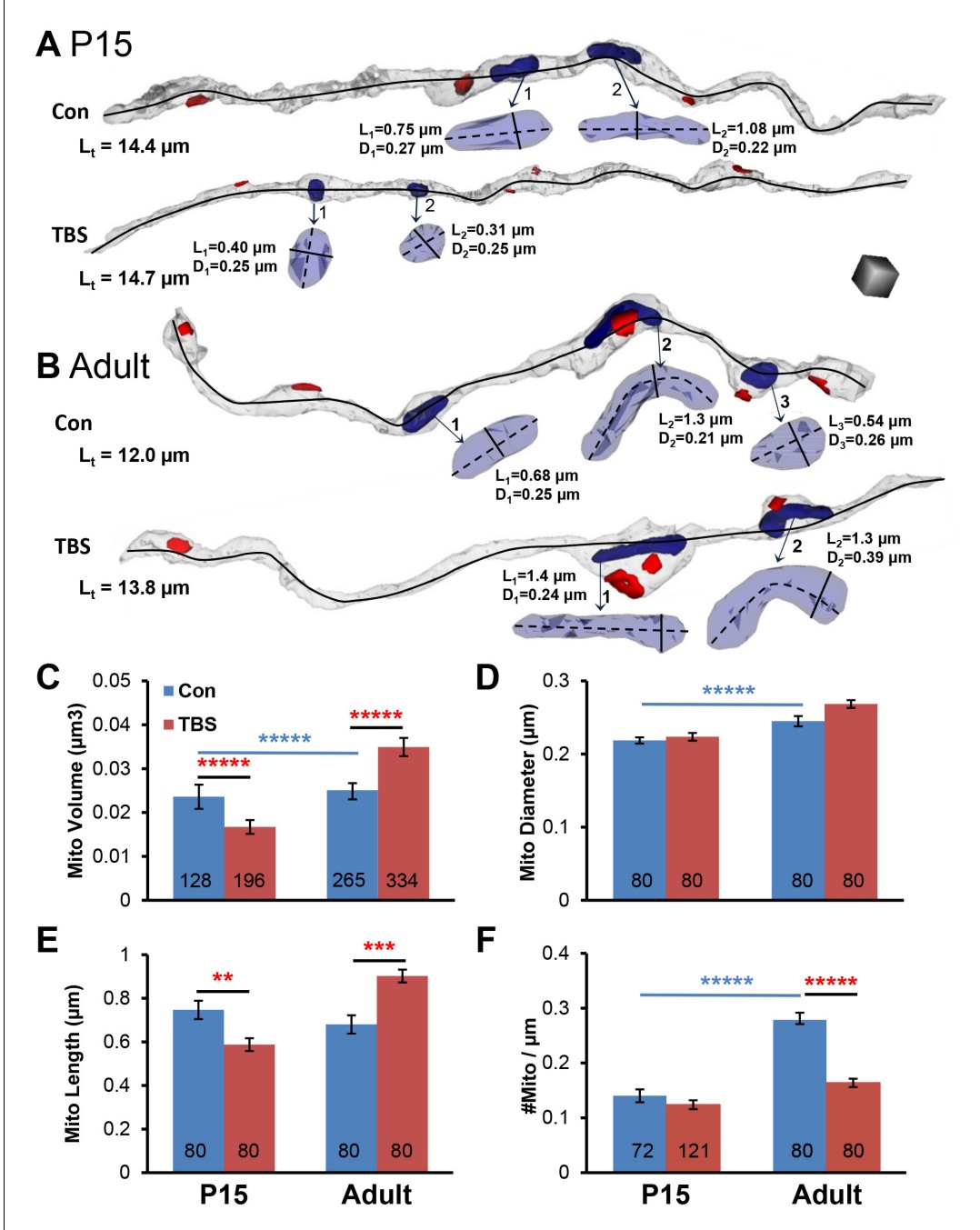

**Figure 9.** Changes in mitochondrial dimensions in P15 and adult hippocampal axons. (**A–B**) Representative 3D reconstructions of axons from P15 and adult (Adult) control and TBS conditions at 120 min. (Total axon segment lengths – $L_t$, black lines – mitochondrial dimensions, length – L, diameter – D, and scale cube is 0.5 µm per side.) (**C**) Mitochondrial volumes were smaller at P15 and larger in adults in the 120 min TBS vs. the control condition ($F_{1,919} = 74.4$, p<0.001, n = all mitochondria for all measured axons). (**D**) Neither P15 nor adult mitochondria showed differences in diameter between conditions (n = 320, $F_{1,316} = 0.37$, p=0.55), although adult mitochondria were wider than P15 mitochondria (n = 320, $F_{1,316} = 37.5$, p<0.001). (**E**) P15 mitochondria were shorter and adult mitochondria were longer at 120 min after TBS vs. the control condition (n = 320, $F_{1,316} = 30.0$, p<0.001). (**F**) Mitochondria per axon segment length (#mito/µm) was greater in adults than P15 in the control condition ($F_{1,349} = 73.2$, p<0.001, blue bar and stars), and the adult mitochondria frequency was significantly reduced in the 120 min post TBS versus control condition ($F_{1,349} = 45.6$, p<0.001). (Numbers at the base of each bar indicate the total number of mitochondria in each group. All the mitochondrial volumes were measured. See *Table 1* and Materials and methods for sampling details.).

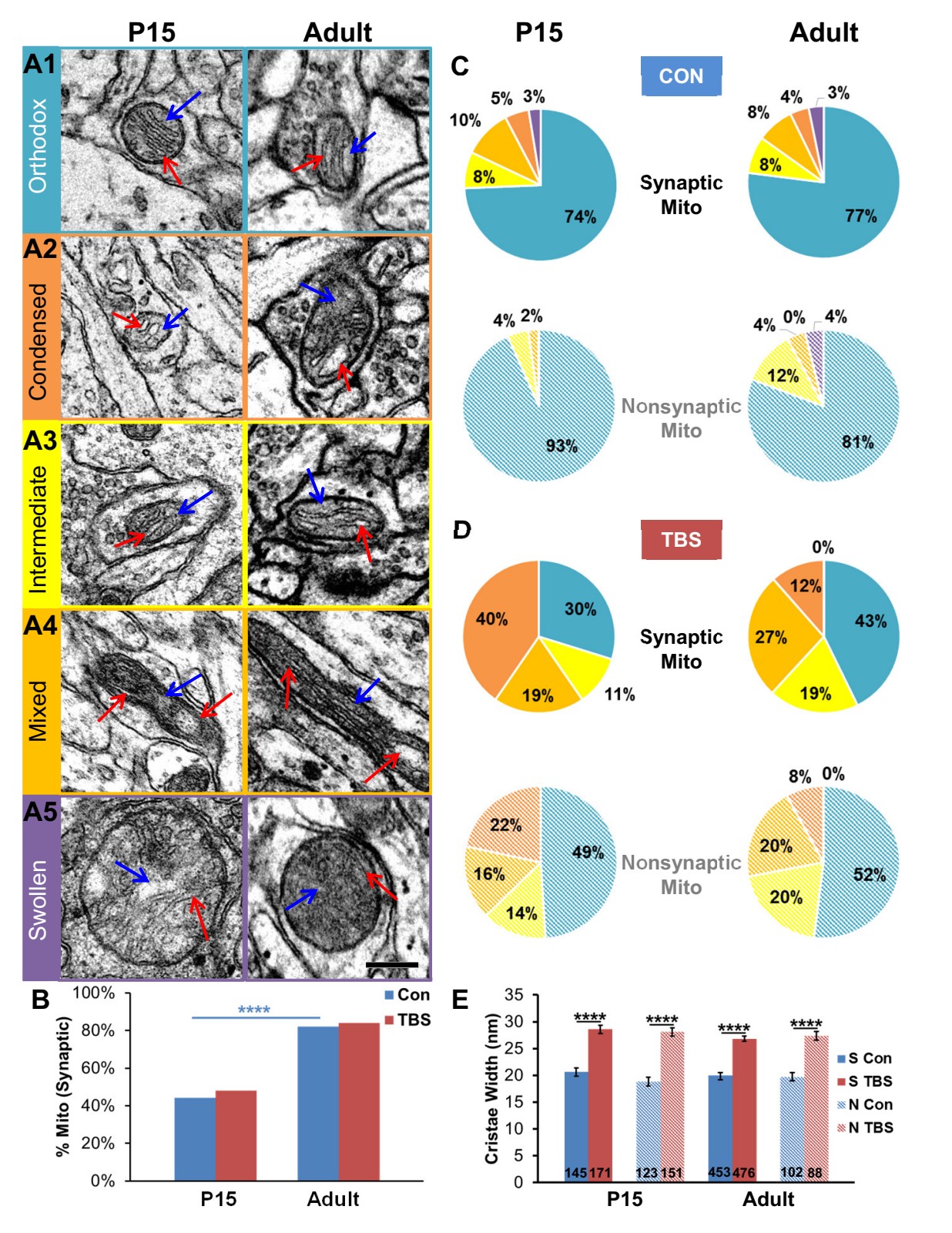

**Figure 10.** Synaptic and nonsynaptic mitochondria exhibited non-orthodox configurations after TBS-induction of LTP at both ages. (A1) The orthodox mitochondrial configuration had cristae with uniformly thin widths (<20 nm) and homogeneously gray matrices. (A2) The condensed mitochondrial configuration had wider and more variable cristae (>30 nm, red arrows) and less uniform matrices. (A3) Mitochondria with an intermediate configuration had cristae ranging in width from 20 to 30 nm and nonuniform matrices. (A4) Long mitochondria often had a mixed appearance showing distinct

*Figure 10 continued on next page*

*Figure 10 continued*

regions of orthodox, condensed, or intermediate configurations along their lengths. (A5) Swollen mitochondria were more than two times wider than other mitochondria. (Cristae are labeled with red arrows and the matrix is labeled with blue arrows in the micrographs of A1-5. Scale bar in **A5** is 0.25 μm for A1-5.) (**B**) Under control conditions, 44% of P15 mitochondria and 82% of adult mitochondria were synaptic and located among the presynaptic vesicles ($X^2$ = 37.5, df = 1, p<0.0001). These proportions did not differ significantly between TBS and control conditions at either age. (**C**) Comparison of mitochondrial configurations showed no significant differences between synaptic and nonsynaptic mitochondria or between P15 and adult ages under control conditions, or in the (**D**) TBS condition. Significant differences between Con and TBS are discussed in the text. (Pie chart colors in **C** and **D** match colors of the mitochondrial configuration boxes in **A**, with solid shades for synaptic and hatched shades for nonsynaptic mitochondria.) (**E**) At both ages, the cristae were wider in the TBS condition for both synaptic and nonsynaptic mitochondria.

Closer proximity of axonal mitochondria was associated with larger PSD areas having more docked and nondocked presynaptic vesicles at both P15 and adult ages. At P15, half of all axonal mitochondria resided in nonsynaptic inter-bouton regions and only 28% of axonal volume surrounded the mitochondria. Thus, although mitochondria were found all along the axons, synaptic boutons with mitochondria had more vesicles and larger synapses than those without mitochondria. These findings are in agreement with those from cultured hippocampal neurons (*Li et al., 2008*; *Obashi and Okabe, 2013*; *Sapir et al., 2012*) and adult amygdala (*Ostroff et al., 2012*), where presynaptic boutons with mitochondria also had more vesicles. Only the presynaptic boutons that contained mitochondria showed a decrease in docked and nondocked vesicles after TBS-LTP at P15. In contrast, the sustained mobilization of vesicles after TBS-LTP was not as strictly dependent on mitochondrial proximity in adults. Results from cultured hippocampal neurons using FM1-43 show that a large proportion of axonal boutons do not release synaptic vesicles (*Crawford and Mennerick, 2012*). Chemical induction of LTP produces an increase in the number of release sites without enhancing the average vesicle release probability (*Bolshakov et al., 1997*), suggesting that the new synapses in this preparation might also be presynaptically silent. New synapses lack postsynaptic AMPA-type glutamate receptors (*Petralia et al., 1999*; *Shi et al., 1999*) and hence also could be postsynaptically silent.

Consistent with our findings, a decrease in the number of synaptic vesicles following LTP has been reported for nearly forty years (*Applegate et al., 1987*; *Fifková and Van Harreveld, 1977*; *Lushnikova, et al., 2008*; *Bourne et al., 2013*; *Bayazitov et al., 2007*; *Stanton et al., 2005*), although not in the context of presynaptic mitochondria, postsynaptic growth, or explicit comparison between immature and mature synapses as reported here. The pools of presynaptic vesicles are now known to differ among those engaged in spontaneous versus evoked release (*Kavalali, 2015*; *Sara et al., 2005*, *2011*; *Espinosa and Kavalali, 2009*; *Sutton et al., 2006*, *2007*; *Sutton and Schuman, 2009*). Enhanced mobilization of either pool could result in a decrease in presynaptic vesicles. Following induction of LTP, both evoked vesicle release probability (*Enoki et al., 2009*; *Stanton et al., 2005*; *Zakharenko et al., 2001*) and spontaneous miniature excitatory synaptic potentials (*Malgaroli and Tsien, 1992*; *1995*; *Manabe et al., 1992*; *Fernández-Ruiz et al., 2012*) have been reported to be increased. Since the potentiation of the (evoked) fEPSP plateaued by 5 min, but TBS-related vesicle mobilization was only detected after 30 min, the persistent enhancement of vesicle mobilization seems more likely to have involved spontaneously released vesicles.

The absence of fission or changes in position suggests that resident synaptic mitochondria were not highly motile even 120 min after TBS at either age. These results are consistent with live imaging in cultured neurons where most axonal mitochondria remained stationary at synapses for hours to days (*Obashi and Okabe, 2013*). As such, boutons with mitochondria will have greater access to the products and functions of mitochondria. Previous findings suggest that persistent vesicular mobilization puts a high demand on mitochondrial production of ATP (*Vos et al., 2010*; *Rangaraju et al., 2014*). Indeed, ATP binding by synapsin IIa regulates the recruitment of reserve pool vesicles into the more readily releasable pool (*Shulman et al., 2015*). Mitochondrial ATP production is also coupled to synaptic vesicle recycling (*Rangaraju et al., 2014*). The decrease in the orthodox configuration and widening of mitochondrial cristae provided evidence for an increase in demand for ATP lasting at least 120 min after TBS-LTP (*Hackenbrock, 1966*; *Packer, 1960*).

Presynaptic mitochondria also serve to regulate calcium. Mitochondrial calcium buffering enhances short-term plasticity, such as post-tetanic potentiation (*Billups and Forsythe, 2002*; *Tang and*

*Zucker, 1997*; *Lee et al., 2007*). Mitochondria are required for vesicular release during high frequency stimulation of *Drosophila* neuromuscular junctions (*Verstreken et al., 2005*) and cultured hippocampal neurons (*Sun et al., 2013*). However, blocking mitochondrial calcium buffering had no effect on synaptic vesicle release in hippocampal neurons and instead mitochondrial ATP was required (*Sun et al., 2013*). Hence, the demand for ATP, rather than calcium sequestration and release, appears to account for the effect of mitochondrial proximity on enhanced vesicular mobilization following TBS-LTP.

Developmental changes in glucose metabolism could explain why only those presynaptic boutons that contained mitochondria showed increased vesicular mobilization at P15 but some presynaptic boutons without mitochondria also had increased vesicle mobilization in adults. Before weaning, rat diets consist primarily of fatty acids, which means their neurons rely on ketone bodies instead of glucose for energy (*Edmond et al., 1985*; *Raffo et al., 2004*; *Yeh and Sheehan, 1985*). Accordingly, rat hippocampal neurons begin to express glucose transporters at about P14, but expression does not approach adult levels until after P21 (*Vannucci, 1994*). Ketone body metabolism requires mitochondria while glucose metabolism does not (*Fukao et al., 2004*). Previous work in cultured hippocampal neurons demonstrates that both mitochondrial and glycolytic ATP production are upregulated under conditions of increased vesicle turnover (*Rangaraju et al., 2014*). Thus, in adults, increased glycolysis could supply ATP in boutons that lack mitochondria. Additionally, the evidence for mitochondrial fusion in the adults suggests that mechanisms to replenish mitochondrial resources were triggered in the adults, consistent with prior work in older animals (*Huang et al., 2015*; *Wang et al., 2009*).

The sustained decrease in presynaptic vesicles could also reflect their use in the growth of active zones and plasma membranes surrounding the presynaptic bouton. Engaging mechanisms for vesicles to 'release and stay', rather than recycle, would serve to enlarge the presynaptic zone. Consistent with this interpretation, some reserve pool vesicles are known to provide a steady source of proteins to the active zone (*Denker and Rizzoli, 2010*; *Denker et al., 2011*). Brain-derived neurotrophic factor (BDNF) plays a role in LTP (*Zakharenko et al., 2003*; *Kang and Schuman, 1996*, *1995*), increases mitochondrial activity (*Markham et al., 2004*), arrests mitochondrial movement (*Su et al., 2014*), and enhances presynaptic vesicle cycling (*Tyler et al., 2006*). Thus, secretion of BDNF or related processes might link presynaptic mitochondrial activity to a persistent mobilization of presynaptic vesicles and serve to coordinate pre- and postsynaptic changes that enhance connectivity at P15 and balance synaptogenesis and growth in adults.

Spontaneous release normally suppresses local protein synthesis in dendrites (*Sutton et al., 2007*). The positions of postsynaptic polyribosomes (the structural correlates of local protein synthesis) are dynamic following TBS-LTP in adults (*Bourne and Harris, 2011*). Polyribosomes are immediately recruited to large spine heads at 5 min. Between 5 and 30 min, polyribosomes become further elevated throughout the dendrites and spines under both control and TBS-LTP conditions. These dynamics could reflect protein synthesis-dependent processes for both control spinogenesis and TBS-LTP-related synapse enlargement by 120 min in adults. Similar analyses are not available for TBS-LTP at P15, although polyribosomes are known to be dynamic following tetanus-induced LTP (*Ostroff et al., 2002*). Thus, if the presynaptic vesicle decrease resulted from increased spontaneous release, as discussed above, it might account for the return of local postsynaptic translation to control levels once the silent TBS-related synaptogenesis at P15 (or growth in adults) is complete.

Being an essentially silent process raises the question about the function of synaptogenesis or synapse growth following TBS. Despite the age-related differences in the sites of synaptogenesis, the occurrence of synapse enlargement, or the degree of vesicle mobilization, the responses of mitochondrial ultrastructure to TBS were nearly identical. At both ages, there was no change in the percentage of mitochondria that were located at presynaptic versus nonsynaptic sites along the axons. At both ages, more of the mitochondria had a non-orthodox configuration and the cristae were enlarged following TBS-LTP for synaptic and nonsynaptic mitochondria. These observations suggest that there were increased demands for ATP production all along the axons following TBS-LTP. Of course, oxidative stress and calcium concentrations also affect mitochondrial structure (*Frezza et al., 2006*; *Hackenbrock and Caplan, 1969*; *Scalettar et al., 1991*). Future studies directly measuring ATP concentrations after TBS, together with higher resolution analyses of mitochondrial ultrastructure (*Perkins et al., 2010*), will help to elucidate whether the LTP-associated changes in mitochondrial structure are a consequence of metabolic demand. Spinogenesis and synapse growth require

energy and may reflect the preparation of synapses to respond to the next bout of plasticity. At P15, the next bout could stabilize the new connectivity, while in adults the next bout could recruit vesicles to new docking sites at the enlarged nascent zones. This interpretation is consistent with recent findings showing that sufficient time must pass before the initial LTP became unsaturated and a second bout of TBS could augment the potentiation (*Babayan et al., 2012*; *Bell et al., 2014*; *Cao and Harris, 2014*).

## Materials and methods

All animal use procedures were approved by the Institutional Animal Care and Use Committee at The University of Texas at Austin and complied with the NIH requirements for the humane use of laboratory rats.

### Slice preparation, electrophysiology, and microwave fixation

Acute hippocampal slices were prepared from male Long-Evans rats (RRID:RGD_2308852) at postnatal day 15 (P15, n = 7; *Watson et al., 2016*). Results were compared to identically prepared slices from young adult Long-Evans rats (55–70 days old, n = 7, *Bell et al., 2014*; *Bourne and Harris, 2011*; *Bourne et al., 2013*). These are referred to as adult rats throughout this paper. Hippocampal slices (400 µm) were prepared from the left hippocampus (Stoelting Co., Wood Dale, IL), 70° from the long axis, at room temperature (~25°C), and immediately placed in oxygenated, balanced salts with glucose (117 mM NaCl, 5.3 mM KCl, 26 mM NaHCO$_3$, 1 mM NaH$_2$PO$_4$, 2.5 mM CaCl$_2$, 1.3 mM MgSO$_4$, and 10 mM glucose, pH 7.4). Slices were transferred to a static-pool interface chamber on supporting nets at the interface between warmed aCSF (33–34°C) and humidified carbogen (95% O$_2$, 5% CO$_2$) to recover for 2.5 hr.

Two concentric bipolar stimulating electrodes (diameter 100 µm, Fredrick Haer, Brunswick, ME) were lowered into the middle of *stratum radiatum* in area CA1 at a separation of at least 600 µm, which ensured stimulation of independent axon populations (*Sorra and Harris, 1998*; *Ostroff et al., 2002*; *Bourne and Harris, 2011*). A glass recording pipette filled with 120 mM NaCl was placed in between the two stimulating electrodes, approximately 300 µm from each (*Figure 1A*).

Custom-designed Igor software (Wavemetrics, Lake Oswego, OR) was used to administer stimulation protocols and acquire data. Field excitatory post synaptic potentials (fEPSPs) were estimated by linear regression over 400 µs along the maximal initial slope (mV/ms) of the response following the test pulse. Test pulses, which were delivered at a rate of one pulse every two minutes, consisted of 100 µs of constant biphasic current at an intensity level designed to evoke a half-maximal fEPSP and then were held constant for the remainder of the experiment. After recording the responses to test pulse stimulation for forty minutes, theta-burst stimulation (TBS: 8 trains of ten bursts at 5 Hz of four pulses at 100 Hz delivered 30 s apart) was delivered to one of the two stimulating electrodes. Then test pulses were alternated between the TBS and control stimulating electrodes and responses were monitored for 5 min (n = 2), 30 min (n = 3), or 120 min (n = 2) after the onset of TBS (*Figure 1B*, *Watson et al., 2016*). We also compared results from adult hippocampus that were given the same stimulation protocol (*Figure 1C*, *Bourne and Harris, 2011*). At the end of each experiment, slices were rapidly fixed via microwave-enhanced fixation (*Jensen and Harris, 1989*).

### Processing and imaging for 3DEM

On the day following the experiment, the fixed slices were gently removed from their nets with a paintbrush and rinsed in 100 mM cacodylate buffer three times for 10 min each. Slices were then embedded in 7% low melting temperature agarose before being trimmed to the region surrounding the two stimulating electrodes and cut into 70 µm thick slices using a vibraslicer (Leica VT 1000S, Leica, Nusslock, Germany). Once the vibraslices containing indentations from the stimulating electrodes were located, they and the two vibraslices immediately adjacent to them were collected and processed in 1% OsO$_4$ and potassium ferrocyanide for ten minutes. After being rinsed five times in 100 mM cacodylate buffer and two times in water (5 min each), vibraslices were immersed in 1% OsO$_4$ and microwaved (1 min on, 1 min off, 1 min on) twice with a cooling down period in between. Slices were then dehydrated through a series of graded ethanols (50%, 70%, 90%, and 100%) with 2% uranyl acetate and microwaved for 40 s at each concentration before being infiltrated at room

temperature with propylene oxide and LX-112 overnight. After being embedded in LX-112 resin, they were cured for 48 hr at 60°C, according to our standard protocol (*Kuwajima et al., 2013*).

Once the slices were cured, a small trapezoid was trimmed in the middle of *stratum radiatum* approximately 150–200 µm from the air surface within about 100 µm of the stimulating electrode. Series of 100–200 ultrathin sections (setting 45–50 nm) were prepared (Ultracut, Leica, Deerfield, IL) and placed on Pioloform-coated slot grids (Synaptek, Ted Pella) and then post-stained with saturated aqueous uranyl acetate followed by Reynolds lead citrate for 5 min each. Sections were dried overnight and then imaged on a JEOL 1230 transmission electron microscope and photographed with a Gatan camera at 5000X magnification. Each series was assigned a code of five random letters so that data collection and analysis were done blind as to experimental condition.

## Perfusion-fixation procedure

Two P15 male Long-Evans rats were anesthetized with pentobarbital (80 mg/kg) and perfused transcardially with fixative (2.5% glutaraldehyde and 2% paraformaldehyde in 100 mM cacodylate buffer, pH 7.4, with 2 mM $CaCl_2$ and 4 mM $MgSO_4$) at 37°C and 4 psi backing pressure and left undisturbed for at least 1 hr. Whole brains were then removed and placed in the same fixative overnight. Brains were then rinsed in buffer and chopped into 400 µm thick sections from which CA1 was dissected, embedded in agarose, and vibrasliced at 70 µm. These vibraslices were processed for 3DEM as described above for the slices.

## 3DEM data collection and analysis

Images were aligned and 3D reconstructions were created using Reconstruct software (http://synapseweb.clm.utexas.edu/software-0). Each coded series of electron micrographs was imported into Reconstruct with a diffraction grating replica (Ernest Fullam Inc, Latham, NY) for pixel size calibration. Images were manually aligned, and section thickness was calibrated using the cylindrical diameters method (*Fiala and Harris, 2001a*): the diameters of longitudinally sectioned mitochondria were measured and compared to the number of sections they traversed. Average section thicknesses ranged from 43 to 63 nm across conditions and ages. Prior comparisons demonstrated no significant effect on outcomes due to this variation in section thickness (*Watson et al., 2016*).

### Unbiased data sampling

The data samples were obtained using the unbiased brick and unbiased dendritic segment approaches (*Fiala and Harris, 2001b*). The figure legends discuss which strategies were used for each analysis and *Table 1* provides the sample numbers across conditions.

For the *unbiased brick analyses*, two independent 3.5 × 3.5 µm sampling frames were placed over 50 serial sections on each series and total brick volumes were computed by multiplying section thickness (which varies by series) and summing across series in each condition (*Table 1*). Each sampling frame consisted of two red exclusion lines and two green inclusion lines (*Fiala and Harris, 2001b*). Boutons that touched the exclusion lines or the last section were excluded, thus preventing double-counting and providing an unbiased sample. Bouton frequencies were calculated by dividing the total number of boutons from each category by the brick volume.

The analyses based on synapses from *unbiased dendrite samples* included the presynaptic partners of caliber-matched dendrites that had been analyzed in previous studies (*Bell et al., 2014*; *Bourne et al., 2013*; *Watson et al., 2016*). The dendrites were matched according to the average number of microtubules in the segment, because microtubule count correlates with dendrite cross-sectional area or diameter (*Fiala et al., 2003*). Microtubule counts ranged from 5 to 25 at P15 and 9 to 22 in adults for comparable calibers, and this variance in microtubules number had no significant effect on the results (*Bourne and Harris, 2011*; *Watson et al., 2016*).

### Synaptic area measurements

Postsynaptic density (PSD) areas were measured according to the orientation in which they were sectioned (*Harris et al., 2015*). Perfectly cross-sectioned synapses had distinct pre- and postsynaptic membranes, clefts, and docked vesicles, and their areas were calculated by summing the product of PSD length and section thickness for each section spanned. En face synapses were cut parallel to the PSD surface, appeared in one section, and were measured as the enclosed area on that section.

Obliquely sectioned PSDs were measured as the sum of the total cross-sectioned areas and total en face areas without overlap on adjacent sections.

For boutons with more than one synapse, the individual PSD areas were measured and summed ($\sum$PSD) for the total area per bouton. Boutons were also classified as excitatory or inhibitory by the morphology of their PSDs and vesicles. Asymmetric, Gray's type I synapses (characterized by round presynaptic vesicles with distinct postsynaptic thickenings) were presumed to be excitatory. Symmetric, Gray's type II synapses had narrow postsynaptic thickenings and smaller, pleomorphic or flattened vesicles. Symmetric synapses were infrequent in s. radiatum and thus were excluded from the analyses reported here. The rare axonal growth cones were also excluded.

## Synaptic vesicle counts and presynaptic mitochondrial distance

Small synaptic vesicles were counted if they appeared on just one section, had a visible membrane and slightly gray interior, and were ~45 nm diameter so as to distinguish them from other organelles such as endosomal vesicles or cross-sectioned tubules of smooth endoplasmic reticulum. Docked vesicles, which correlate well with the readily releasable pool (*Dobrunz, 2002*), contacted the presynaptic membrane immediately adjacent to the PSD. Docked vesicles were only counted at cross-sectioned synapses and were analyzed separately from nondocked vesicles. The z-trace tool in Reconstruct was used to measure across serial sections the axon lengths and distances from each PSD to the nearest edge of the nearest mitochondrion.

## Computing axonal volumes with and without mitochondria

The axon volumes were computed by summing the areas inside axon plasma membrane traces and multiplying by section thickness. The axon volumes for mitochondria-containing and mitochondria-free regions were computed separately based on axon traces that either overlapped or had no mitochondria, respectively. Reconstructed axons that had no mitochondria were excluded from this analysis.

## Mitochondrial and axonal dimensions

The z-trace tool was used to measure the diameters and lengths of mitochondria in three dimensions. Mitochondrial lengths were measured by placing a line down the center of the long axis. Diameters were measured by placing a line perpendicular to the length across the widest portion of the mitochondrion. Axonal diameters were measured by placing a z-trace in the same orientation and location as mitochondrial diameters. The diameters of mitochondrial cristae were measured in several (at least five) locations throughout the sampled mitochondria.

## Statistical analyses

Statistical analyses were performed with STATISTICA (Statsoft, Tulsa, OK) and Microsoft Excel. Differences in the volume density of boutons were evaluated using analyses of variance (ANOVA, condition x animal, *Figure 2*). Two-way ANOVAs were used to evaluate differences in the numbers of vesicles and total PSD area per bouton (with the type of bouton, single synaptic or multisynaptic, as one factor and the presence or absence of a mitochondrion as the other factor, *Figure 3*). No significant effects could be attributed to the number of synapses on a presynaptic bouton; instead, total PSD area per bouton carried the effects.

Piecewise regression analyses were performed to determine the threshold distance of synapses from presynaptic mitochondria ($D_m$) where structural differences in vesicle numbers and synapse size became uncorrelated with mitochondrial proximity (*Figure 4*). These analyses were performed using an iterative procedure called the Levenberg-Marquardt algorithm that combines the Gauss-Newton algorithm with the steepest-descent approach to solve nonlinear least-squares problems (*More, 1978*). We began with this general model:

$$y = a1 * x + a2 * (x - c) * (x) + b1$$
$$y(x \leq c) = a1 * D_m + b1$$
$$y(x) = (a1 + a2) * D_m + (b1 - a2 * c),$$

where y was the number of nondocked or docked vesicles or total PSD area at a particular presynaptic bouton. The value of x was $D_m$ for each y, and c was the value of $D_m$ that constituted a

**Table 1.** Sample sizes for axons, boutons, vesicles, and mitochondria under each condition. **Table 1** provides total n's for each of the figures. Dashes mean that category was not used in a particular figure. **Figure 2**: Number of boutons entering the inclusion volume of the unbiased bricks that were identified as single, multi-, or nonsynaptic boutons. **Figures 3–4**: Number of axon segments traced from the presynaptic boutons of dendritic spines on dendrites that were analyzed in **Watson et al. (2016)**. Vesicles were counted and mitochondrial distances measured for all of the boutons along these axons as long as the boutons were complete and nearer to a mitochondrion than to the edge of the image stack. Axons in parentheses contained mitochondria and were used for **Figure 3G**.

**Figures 5–7**: Boutons were the presynaptic partners of the dendrites analyzed in **Watson et al. (2016)**. All boutons were used except for those that were incomplete or fewer than three microns from the edge of the image stack. **Figure 8**: All boutons from **Bourne et al. (2013)** were used except for those that were less than three microns from the edge of an image stack. Two additional bricks were analyzed from each dataset of the adults, for a total of six bricks in each condition for the adults. **Figure 9**: Axon lengths are μm (mean ± sd). **Figure 9C**: Every complete mitochondrion encountered along every axon from the 2-hr P15 and adult series was traced to compute volumes. **Figure 9D and E**: Every fourth mitochondrion was measured to obtain the lengths and diameters for a subset of these mitochondria. A total of 10 presynaptic mitochondria were traced from each series, for a total of 80 mitochondria per condition. **Figure 9F**: Mitochondria frequencies were determined for all axon segments greater than 10 microns long at P15. To obtain comparable measurements in the adults, mitochondria frequencies were measured from 10 axon segments that were at least 10 microns from each series for a total of 80 axons. **Figure 10**: Morphological conformations were identified for all presynaptic mitochondria that fell within the original unbiased bricks at both ages for a total of 4 per condition at P15 and in adults (used the four original bricks from **Bell et al., 2014**). Mitochondria were identified within the sample volumes beginning on the first section of each of the sample volumes used and numbered in the order in which they were found. The widths of each of their cristae were measured for every third mitochondrion at P15 and every fourth mitochondrion in adults to obtain comparable sample sizes.

| Age | P15 | | | | | | | | Adult | |
|---|---|---|---|---|---|---|---|---|---|---|
| Time | Perfused | | 5 min | | 30 min | | 120 min | | 120 min | |
| Condition | R88 | R89 | CON | LTP | CON | LTP | CON | LTP | CON | LTP |
| # dendrites | 9 | 8 | 19 | 20 | 23 | 28 | 24 | 22 | 8 | 8 |
| # bricks | 2 | 2 | – | – | – | – | 4 | 4 | 6, 4 | 6, 4 |
| Total Brick Volume (μm$^2$) | 54 | 65 | – | – | – | - | 147 | 125 | 165, 99 | 222, 145 |
| **Axons (total n)** | | | | | | | | | | |
| *Figure 3, 4* | 85 (44) | 77 (37) | – | - | – | – | – | – | – | – |
| *Figure 5, 6* | – | – | 132 | 108 | 143 | 184 | 88 | 144 | – | – |
| *Figure 7* | – | – | – | – | 143 | 184 | 88 | 144 | – | - |
| *Figure 8* | – | - | – | – | – | – | – | – | 225 | 289 |
| *Figure 9* | – | – | – | – | – | – | 71 | 121 | 80 | 80 |
| *Figure 9* axon length | – | – | – | – | – | – | 12.5 ± 2.9 | 13.0 ± 2.9 | 11.0 ± 1.4 | 12.6 ± 1.7 |
| **Boutons (total n)** | | | | | | | | | | |
| *Figure 2* | 89 | 84 | – | – | – | – | 165 | 208 | – | – |
| *Figure 3, 4* | 77 | 113 | – | – | – | – | – | – | – | – |
| *Figure 5, 6* | – | – | 132 | 108 | 143 | 184 | 88 | 144 | – | – |
| *Figure 7* | – | – | – | – | 143 | 184 | 88 | 144 | – | - |
| *Figure 8* | – | - | – | - | – | - | – | - | 225 | 289 |
| **Mitochondria (total n)** | | | | | | | | | | |
| *Figure 3H,I* | 36 | 39 | – | - | – | – | – | – | – | – |
| *Figure 9C* | – | – | – | - | – | - | 128 | 196 | 265 | 334 |
| *Figure 9D,E* | – | - | – | - | – | - | 80 | 80 | 80 | 80 |
| *Figure 9F* | – | – | – | – | – | – | 71 | 121 | 80 | 80 |
| *Figure 10A–C* | – | – | – | – | – | – | 88 | 105 | 161 | 159 |
| *Figure 10D* | – | – | – | – | – | – | 33 | 36 | 40 | 40 |

breakpoint. The expression (x > c) is a logical operator that becomes zero when the statement is false and one when it is true. The parameters a1 and a2 were constants for the slopes on either side of the breakpoint that were computed by STATISTICA during each iteration. The intercept to the left of the breakpoint (x ≤ c) is with the y axis at b1, and to the right of the breakpoint (x > c) it is (b1 – a2*c) to ensure that the two lines converge at the breakpoint.

Categorical comparisons of boutons and mitochondria were performed using the chi-square test (*Figure 5*, with percentages based on numbers in *Table 1*). The hierarchical nested ANOVA (hnANOVA) was used to evaluate differences in the frequency of docked or nondocked vesicles in boutons with or without mitochondria (*Figure 6*, nested by experiment and condition). Analyses of covariance (ANCOVA) were done to test for differences between control and TBS conditions across PSD areas (*Figures 7* and *8*). The hnANOVA was also used to evaluate differences in mitochondria structure and frequency between control and TBS conditions and for P15 and adult ages (*Figure 9*, with experimental condition nested by animal). Categorical comparisons of mitochondria were performed using the chi-square test (*Figure 10B–D*, with percentages based on numbers in *Table 1*). Cristae widths were compared using hnANOVA (*Figure 10E*, with experimental condition nested by animal).

Vesicle counts, mitochondrial dimensions, and PSD areas have skewed distributions with different variances and thus were transformed using the natural logarithmic function for all of the ANOVAs. Post-hoc analyses were performed using Tukey's HSD test. The variances were similar across conditions, hence no transformations were required for the ANCOVAs. Data were plotted as means with standard error bars or as the raw data points on log axes to visualize both the skewed nature of the raw data and the linear relationships using a Power Regression to generate the trend lines (MS Excel). Synapses with zero docked presynaptic vesicles were set to a value of 0.1 and plotted along the x-axes of the log plots (set at the y value of 0.1). The percentages in the pie charts were based on the relevant numbers in *Table 1*. Results of the statistical analyses are designated in the text or figure legends. Significant post hoc values are indicated on the graphs with stars for *p<0.05, **p<0.01, ***p<0.001, and ****p<0.0001.

## Acknowledgements

We thank Libby Perry, Robert Smith, and John Mendenhall for technical assistance with the electron microscopy. JB, GC, MC, LO, and DW are listed alphabetically by last name as all contributed important components of roughly equal value. HS, JB, DW, and MC performed 3D reconstructions; HS, JB, LO, GC, and KH designed the experiments; JB (P50-75), LO (P15), and GC (P15) performed the slice electrophysiology; KH performed the in vivo perfusions; HS and KH performed the statistical analyses and wrote the paper. We thank members of the Harris Laboratory for insightful and editorial input.

## Additional information

### Funding

| Funder | Grant reference number | Author |
|---|---|---|
| National Institutes of Health | NS201184 | Kristen M Harris |
| Texas Emerging Technology Fund | | Kristen M Harris |
| National Institutes of Health | MH095980 | Kristen M Harris |
| National Institutes of Health | NS074644 | Kristen M Harris |
| National Institutes of Health | MH096459 | Deborah J Watson |
| Brain Research Foundation | | Kristen M Harris |

The funders had no role in study design, data collection and interpretation, or the decision to submit the work for publication.

## Author contributions

HLS, Conception and design, Acquisition of data, Analysis and interpretation of data, Drafting or revising the article; JNB, GC, Conception and design, Acquisition of data, Analysis and interpretation of data; MAC, DJW, Acquisition of data, Analysis and interpretation of data ; LEO, Conception and design, Acquisition of data ; KMH, Conception and design, Acquisition of data, Analysis and interpretation of data, Drafting or revising the article

## Author ORCIDs

Guan Cao, http://orcid.org/0000-0001-6211-5872
Kristen M Harris, http://orcid.org/0000-0002-1943-4744

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
