## [Decision Letter]

Thank you for submitting your article "Mitochondria support persistent presynaptic vesicle mobilization with age-dependent synaptic growth after LTP" for consideration by *eLife*. Your article has been favorably evaluated by Eve Marder as the Senior Editor and three reviewers, one of whom is a member of our Board of Reviewing Editors. The reviewers have opted to remain anonymous.

The reviewers have discussed the reviews with one another and the Reviewing Editor has drafted this decision to help you prepare a revised submission.

Summary:

The authors show that presynaptic mitochondria sustain vesicle mobilization following theta burst stimulation, an LTP protocol, in hippocampal area CA1. This suggests that the presence of mitochondria at a presynaptic bouton has direct consequences for the types of plasticity which this synapse can undergo. It also suggests that the availability of energy (e.g. in the form of ATP) is an important constraint not only for the ongoing operation of synapses, but also their plasticity.

The present study uses wild-type animals (unlike, for example, a previous study by Verstreken et al. 2005, which used *Drosophila* mutants), and thus normal properties and a normal spatial distribution of mitochondria across presynaptic boutons can be assumed to exist at the beginning of the experiments. The authors also compare boutons containing mitochondria, boutons closer than 3 microns from a mitochondrion, and the remaining boutons, providing a direct comparison and internal control under otherwise identical experimental conditions. These methodological choices, together with their use of 3D electron microscopy – the gold standard for structural studies of synapses – has resulted in a compelling and high-quality manuscript. While the approach is essentially descriptive and correlative, the data on vesicle numbers and the correlation with mitochondrial location and volume are particularly compelling and interesting. The twist of analyzing this for young animals, before weaning (P15) vs. adult represents a very neat way to get at morphological correlates for tolerance to repetitive activation.

Although overall the work was judged to be very solid and a pleasure to read, the referees felt that some of the conclusions may have been overstated, and that the authors may have overgeneralized from their paradigm. These issues, and other concerns below, need to be addressed in the revised version.

Essential revisions:

1) While the amount and level of analysis of 3D EM deployed in this work is impressive, the descriptive nature of this work results in overinterpretations, and in statements that are not tested further but are posited as truths. For example, in the last paragraph of the subsection “Mitochondria-related differences in synapse size and vesicle composition at P15 in vivo” they write that the proximity of presynaptic mitochondria supports larger synapses with more vesicles. Maybe the larger synapses are able to accommodate a mitochondrion while the smaller ones do not because of space limitations? It is not clear what is cause and what is consequence. In the last paragraph of the subsection “Influence of distance from mitochondria on synapse size and vesicle content” they then conclude that boutons containing a mitochondrion have more vesicles. If terminals with a mitochondrion are larger there is more space to accommodate vesicles. Hence, the presence of more vesicles is not necessarily due to the presence of a mitochondrion but maybe because of the larger size? In the same vein, it is unclear why the data were not normalized to bouton volume. In the aforementioned paragraph the authors conclude that the presence (or proximity) of a mitochondrion has a positive influence on synaptic efficacy; but this measure of efficacy is simply based on the number of vesicles (and PSD area). Functional studies are needed to make such statements. These issues should be addressed by a careful rewriting of the claims made from the data.

2) Another example of overstating conclusions is in the first paragraph of the subsection “Presynaptic Mitochondria also Sustain Vesicle Mobilization following TBS in Adults”: the authors conclude that the vesicle mobilization in adult synapses without mitochondria appears to be supported by glycolysis but no proof to support this statement that glycolysis supports the mobilization in these boutons is included. Further down they analyze the morphology of the cristae of the mitochondria and use these results to make statements about ATP levels and production. (e.g.: “these changes in mitochondria structure indicate an ongoing elevation in demand for presynaptic ATP at 120 min after TBS-LTP”). This is at best correlative and it is likely that other aspects of mitochondrial biology affect cristae structure as well. There are methods to measure ATP (also in situ) that could be used rather than having to rely on proxies to make conclusions that are unfounded. Again, careful rewriting of these sections should be performed to address these issues.

3) Statements like "Thus, the proximity of presynaptic mitochondria supports larger synapses with more vesicles" and "These results suggest that the proximity of a mitochondrion has a strongly positive influence on synaptic efficacy" should be qualified more carefully since cause and effect might also work in the opposite direction. Consider the following hypothesis: there is a certain average density of mitochondria per axon volume, but mitochondria are randomly distributed in this volume, and larger boutons therefore have a larger chance of containing a mitochondrion. We know from past studies that larger (by volume) boutons also tend to contain more vesicles, contain larger and more active zones, and form stronger synapses. How about this theory as the explanation for the results shown in Figure 3 and Figure 4? The authors should compare their data in Figure 3 and Figure 4 with the predictions of a simple model of random placement of mitochondria inside the volume of their axonal reconstructions, assuming a constant mean density given by the total number of mitochondria divided by the sum of the volumes of the axons they reconstructed.

4) Relating specifically to Figure 3: what is the relative volume of SSBs and MSBs? Could the increased fraction of MSBs containing a mitochondrion simply be a consequence of a larger average volume of MSBs compared to SSBs (assuming a constant mean density of mitochondria per volume in the axon, and random placement of mitochondria according to this density)? Even if not, which percentage of the SSB-MSB effect shown in Figure 3 could simply be explained by random placement of mitochondria in the axon volume?

5) Figure 8: in adult hippocampus, TBS leads to vesicle mobilization also in boutons without mitochondria (albeit to a lesser extent than in boutons with mitochondria). The importance of the presence of mitochondria for vesicle mobilization in adult synapses is therefore less clear. Is there a better way of quantifying the size of the TBS-control difference between Figure 8 than the p-values, which are the same order of magnitude?

6) Why does the density of mitochondria per axon length drop to almost half of its control value after TBS (Figure 9, Adult) while the fraction of boutons containing a mitochondrion, and the fraction of boutons within 3 µm of a mitochondrion, stay almost the same (compare Figure 8)? One possible conclusion is that the (smaller) adult axon population analyzed in Figure 9 might not be representative of the (larger) adult axon population analyzed in Figure 8. In any case, the authors should find the reason(s) for, and address, this apparent inconsistency.

7) Although only a small quibble, data, regarding the structural state of the mitochondria – based on cristae widths and matrix structure, are weaker than other data in the manuscript. Their strategy is consistent with the way these differences are described in the literature, (orthodox vs. condensed state). Their analysis shows significant differences for the parameters measured, however the resolution and morphology of the images shown to represent the data analyzed do not engender confidence in data pertaining to those measures. 3D analysis of mitochondria (e.g., by EM Tomography, ssTEM, SBEM or FIB-SEM) to assess the overall pattern of the cristae and their subcomponent volume fractions may have served this purpose better. Lammelar cristae are generally associated with stationary mitochondria at node of Ranvier or in terminals while elongated mitochondria in axons tend to have longitudinally oriented cristae. Although difficult to see in Figure 10, it looks like they have both types in their terminals and analysis.

[Editors' note: further revisions were requested prior to acceptance, as described below.]

Thank you for resubmitting your work entitled "Mitochondrial support of persistent presynaptic vesicle mobilization with age-dependent synaptic growth after LTP" for further consideration at *eLife*. Your revised article has been favorably evaluated by Eve Marder as the Senior Editor and a Reviewing editor.

The manuscript has been improved but there are some remaining issues that need to be addressed before acceptance, as outlined below.

The authors have satisfactorily addressed most of the comments of the reviewers, and have significantly improved the manuscript. However, one issue, which was picked up by both reviewers, has not yet been fully addressed. In their rebuttal, the authors write:

"The new analyses added to Figure 3 and discussed in item 1 above also address the theory offered about potential mitochondrial distribution being random vs. nonrandom. Indeed, as now shown in Figure 3, mitochondria are distributed roughly equally among synaptic and nonsynaptic locations. Furthermore, both synaptic and nonsynaptic mitochondria would easily fit in places where axons are at their thinnest, in the inter-bouton regions. This analysis clarifies that bouton or axon size does not restrict mitochondria locations."

The new data presented in Figure 3 are a step in the right direction, but unfortunately, they do not yet address whether mitochondria are distributed randomly in the axonal volume available to them.

To assess whether the 50.7% of mitochondria located in inter-bouton regions and the 49.3% located in synaptic boutons represent a random distribution in space, these percentages of mitochondria need to be related to the percentages of the volumes of the inter-bouton regions and the synaptic boutons. For example, if these volume percentages were similar to those in adult axons, then it would be clear that mitochondria are non-randomly distributed, given that in adult axons the varicosities constitute 80% of the total volume (Shepherd & Harris 1998 p. 8305; see also Table 3 on p. 8304).

Since the authors have this data also for P15 axons, they should test whether the spatial distribution of mitochondria could simply reflect the volume available to them, or whether the distribution of mitochondria is nonrandom. It would be most interesting – and strengthen the paper – if the authors could demonstrate that the results in the current Figure 3 and Figure 4 cannot simply be explained by larger structures being more likely to contain mitochondria (in a model assuming a random spatial distribution of mitochondria).

---

## [Author Response]

*Essential revisions:*

*1) While the amount and level of analysis of 3D EM deployed in this work is impressive, the descriptive nature of this work results in overinterpretations, and in statements that are not tested further but are posited as truths. For example, in the last paragraph of the subsection “Mitochondria-related differences in synapse size and vesicle composition at P15* in vivo*” they write that the proximity of presynaptic mitochondria supports larger synapses with more vesicles. Maybe the larger synapses are able to accommodate a mitochondrion while the smaller ones do not because of space limitations? It is not clear what is cause and what is consequence. In the last paragraph of the subsection “Influence of distance from mitochondria on synapse size and vesicle content” they then conclude that boutons containing a mitochondrion have more vesicles. If terminals with a mitochondrion are larger there is more space to accommodate vesicles. Hence, the presence of more vesicles is not necessarily due to the presence of a mitochondrion but maybe because of the larger size? In the same vein, it is unclear why the data were not normalized to bouton volume. In the aforementioned paragraph the authors conclude that the presence (or proximity) of a mitochondrion has a positive influence on synaptic efficacy; but this measure of efficacy is simply based on the number of vesicles (and PSD area). Functional studies are needed to make such statements. These issues should be addressed by a careful rewriting of the claims made from the data.*

Thank you for these insights. To address these questions, we have added analyses of the relationships between mitochondrial and axonal dimensions. These analyses are presented in Figure 3 (panel b4), 3C, 3D, and related text and figure legends. Nearly 50% of the mitochondria occurred outside synaptic boutons, in nonsynaptic regions of the axon with no surrounding vesicles. The synaptic and nonsynaptic mitochondria do not differ significantly from one another in their diameters. Both would easily fit even in places where there are no synapses and the axons are at their thinnest, in the inter-bouton regions (e.g. see the nonsynaptic mitochondrion of b4 in Figure 3 (3DEM), and 3B (b4, EM). This analysis clarifies that bouton or axon size does not restrict mitochondria locations. Similarly, there is room for vesicles in the nonsynaptic portions of the axon, as can be seen in all of our 3DEMs, as the vesicles (green) are all included at scale.

We have also modified the text as requested changing synapse “efficacy” to “synapse size and vesicle number”.

*2) Another example of overstating conclusions is in the first paragraph of the subsection “Presynaptic Mitochondria also Sustain Vesicle Mobilization following TBS in Adults”: the authors conclude that the vesicle mobilization in adult synapses without mitochondria appears to be supported by glycolysis but no proof to support this statement that glycolysis supports the mobilization in these boutons is included. Further down they analyze the morphology of the cristae of the mitochondria and use these results to make statements about ATP levels and production. (e.g.: “these changes in mitochondria structure indicate an ongoing elevation in demand for presynaptic ATP at 120 min after TBS-LTP”). This is at best correlative and it is likely that other aspects of mitochondrial biology affect cristae structure as well. There are methods to measure ATP (also in situ) that could be used rather than having to rely on proxies to make conclusions that are unfounded. Again, careful rewriting of these sections should be performed to address these issues.*

We revised the concluding remark about glycolysis as follows:

“Furthermore, in adults, but not at P15, the elevated vesicular mobilization also occurred in boutons without mitochondria, which might have been feasible because glycolysis does not require mitochondria and can be a source of ATP in adults but not at P15 (Vannucci, 1994).”

We revised the statement about ATP demand as follows:

“These changes in mitochondria structure are consistent with an elevated demand for presynaptic ATP at 120 minutes after TBS-LTP.”

*3) Statements like "Thus, the proximity of presynaptic mitochondria supports larger synapses with more vesicles" and "These results suggest that the proximity of a mitochondrion has a strongly positive influence on synaptic efficacy" should be qualified more carefully since cause and effect might also work in the opposite direction. Consider the following hypothesis: there is a certain average density of mitochondria per axon volume, but mitochondria are randomly distributed in this volume, and larger boutons therefore have a larger chance of containing a mitochondrion. We know from past studies that larger (by volume) boutons also tend to contain more vesicles, contain larger and more active zones, and form stronger synapses. How about this theory as the explanation for the results shown in Figure 3 and Figure 4? The authors should compare their data in Figure 3 and Figure 4 with the predictions of a simple model of random placement of mitochondria inside the volume of their axonal reconstructions, assuming a constant mean density given by the total number of mitochondria divided by the sum of the volumes of the axons they reconstructed.*

We thank the reviewer for their thoughtful comments. Similar to item 1 above, we have modified the language throughout the paper to be more circumspect about cause, effect, and function. In addition to the examples indicated above, we changed most statements involving the phrase ‘mitochondria support’ to ‘mitochondria were associated with’ and ‘synaptic efficacy’ to ‘synapse size or vesicle number’ where appropriate throughout the manuscript. We left the word ‘support’ in the title, because that is what we conclude based on our results together with findings from the literature. In deference to the reviewer’s comments, we added the word ‘of’ after the word support to be more circumspect about function.

The new analyses added to Figure 3 and discussed in item 1 above also address the theory offered about potential mitochondrial distribution being random vs nonrandom. Indeed, as now shown in Figure 3, mitochondria are distributed roughly equally among synaptic and nonsynaptic locations. Furthermore, both synaptic and nonsynaptic mitochondria would easily fit in places where axons are at their thinnest, in the inter- bouton regions. This analysis clarifies that bouton or axon size does not restrict mitochondria locations. Figure 3 and Figure 4 are designed to assess whether synapse proximity to presynaptic mitochondria is associated with differences in synapse size and vesicle content. Hence, along with the clarification presented in Figure 3 we have respectfully retained the analyses in Figure 4, as it provides the initial breakdown of effects on synapse parameters with respect to distance from a mitochondrion, used in subsequent analyses.

*4) Relating specifically to Figure 3: what is the relative volume of SSBs and MSBs? Could the increased fraction of MSBs containing a mitochondrion simply be a consequence of a larger average volume of MSBs compared to SSBs (assuming a constant mean density of mitochondria per volume in the axon, and random placement of mitochondria according to this density)? Even if not, which percentage of the SSB-MSB effect shown in Figure 3 could simply be explained by random placement of mitochondria in the axon volume?*

This inquiry is also addressed by the new analyses of axon, bouton, and mitochondrial size and location in revised Figure 3 as discussed above for item 1.

Indeed, MSBs are larger than SSBs, but mitochondria location is not limited by bouton or axon dimensions.

*5) Figure 8: in adult hippocampus, TBS leads to vesicle mobilization also in boutons without mitochondria (albeit to a lesser extent than in boutons with mitochondria). The importance of the presence of mitochondria for vesicle mobilization in adult synapses is therefore less clear. Is there a better way of quantifying the size of the TBS-control difference between Figure 8 than the p-values, which are the same order of magnitude?*

The F statistics are provided in the Figure legend 8. For boutons with mitochondria F=27.2 for the nondocked vesicle number drop with TBS-LTP; whereas, for boutons without mitochondria F=4.72. These F values show the large difference in effect size even though both achieved similar p values.

*6) Why does the density of mitochondria per axon length drop to almost half of its control value after TBS (Figure 9, Adult) while the fraction of boutons containing a mitochondrion, and the fraction of boutons within 3 µm of a mitochondrion, stay almost the same (compare Figure 8)? One possible conclusion is that the (smaller) adult axon population analyzed in Figure 9 might not be representative of the (larger) adult axon population analyzed in Figure 8. In any case, the authors should find the reason(s) for, and address, this apparent inconsistency.*

We thank the reviewer for noticing this apparent discrepancy. In response, we have added a new Figure 8, which illustrates that the density of synaptic boutons was 50% less in the TBS versus the Control conditions at 2 hours after the induction of LTP in the adult hippocampus. This difference matches the 50% difference in synapse number, where small dendritic spines that form in response to control stimulation fail to form following LTP, when instead synapses enlarged (Bourne and Harris, 2011). The data in new Figure 8 are a recapitulation of our published findings (Bourne JN, Chirillo MA, Harris KM. 2013). We agree that it is useful to republish them here to make it clear that the relative fractions of boutons with and without mitochondria were maintained in adults, despite the differences in axonal bouton numbers.

7) Although only a small quibble, data, regarding the structural state of the mitochondria – based on cristae widths and matrix structure, are weaker than other data in the manuscript. Their strategy is consistent with the way these differences are described in the literature, (orthodox vs. condensed state). Their analysis shows significant differences for the parameters measured, however the resolution and morphology of the images shown to represent the data analyzed do not engender confidence in data pertaining to those measures. 3D analysis of mitochondria (e.g., by EM Tomography, ssTEM, SBEM or FIB-SEM) to assess the overall pattern of the cristae and their subcomponent volume fractions may have served this purpose better. Lammelar cristae are generally associated with stationary mitochondria at node of Ranvier or in terminals while elongated mitochondria in axons tend to have longitudinally oriented cristae. Although difficult to see in Figure 10, it looks like they have both types in their terminals and analysis.

We have tempered our text in light of these caveats (e.g. we added “might have”: “The mitochondrial configurations described in these papers and measures of cristae widths were used as proxies to investigate whether changes in energy demand might haveoccurred at 120 min after TBS.”). In addition, we added the following statement, “Future studies directly measuring ATP concentrations after TBS, together with higher resolution analyses of mitochondrial ultrastructure (Perkins et al., 2010), will help to elucidate whether the LTP-associated changes in mitochondrial structure are a consequence of metabolic demand.” We appreciate the importance of the additional study, which is beyond the scope of this work at present. We believe that these data are sufficiently suggestive as to provide motivation for the more demanding analyses in future work. Hence, we have left them in this paper.

[Editors' note: further revisions were requested prior to acceptance, as described below.]

*The manuscript has been improved but there are some remaining issues that need to be addressed before acceptance, as outlined below.*

*The authors have satisfactorily addressed most of the comments of the reviewers, and have significantly improved the manuscript. However, one issue, which was picked up by both reviewers, has not yet been fully addressed. In their rebuttal, the authors write:*

*"The new analyses added to Figure 3 and discussed in item 1 above also address the theory offered about potential mitochondrial distribution being random vs. nonrandom. Indeed, as now shown in Figure 3, mitochondria are distributed roughly equally among synaptic and nonsynaptic locations. Furthermore, both synaptic and nonsynaptic mitochondria would easily fit in places where axons are at their thinnest, in the inter-bouton regions. This analysis clarifies that bouton or axon size does not restrict mitochondria locations."*

*The new data presented in Figure 3 are a step in the right direction, but unfortunately, they do not yet address whether mitochondria are distributed randomly in the axonal volume available to them.*

*To assess whether the 50.7% of mitochondria located in inter-bouton regions and the 49.3% located in synaptic boutons represent a random distribution in space, these percentages of mitochondria need to be related to the percentages of the volumes of the inter-bouton regions and the synaptic boutons. For example, if these volume percentages were similar to those in adult axons, then it would be clear that mitochondria are non-randomly distributed, given that in adult axons the varicosities constitute 80% of the total volume (Shepherd & Harris 1998 p. 8305; see also* Table 3 *on p. 8304).*

*Since the authors have this data also for P15 axons, they should test whether the spatial distribution of mitochondria could simply reflect the volume available to them, or whether the distribution of mitochondria is nonrandom. It would be most interesting – and strengthen the paper – if the authors could demonstrate that the results in the current Figure 3 and Figure 4 cannot simply be explained by larger structures being more likely to contain mitochondria (in a model assuming a random spatial distribution of mitochondria).*

We have further revised Figure 3, adding the graphs of Figure 3, and we have revised the related text in the Methods, Figure Legends, and in the Results (as quoted below) to address this concern. The figure legend for Figure 3 was also substantially modified in the manuscript to include detailed statistical analysis of the new data, and a Methods Section was added to describe exactly how these measurements were made.

From Results:

“These configurations might have resulted from mitochondria being preferentially located in larger axonal swellings. Then their greater association with more vesicles might have been due to chance because the larger synaptic boutons contained more vesicles. […] Thus, the significant effects of mitochondrial proximity on changes in synapse size and presynaptic composition reported below do not appear merely to be due to a chance restriction of mitochondria to larger synaptic boutons.”

In Methods:

“Computing Axonal Volumes with and without Mitochondria: The axon volumes were computed by summing the areas inside axon plasma membrane traces and multiplying by section thickness. The axon volumes for mitochondria-containing and mitochondria-free regions were computed separately based on axon traces that either overlapped or had no mitochondria, respectively. Reconstructed axons that had no mitochondria were excluded from this analysis.”

In Figure 3 portion of the Legend:

“G) Axon segments containing mitochondria somewhere along their lengths were analyzed and the amount of axon volume with mitochondria was significantly less than the amount of axon volume without mitochondria (n=81, ANOVA, F_1,161_=104.1, p<0.0001). […] I) Axonal and mitochondrial diameters were correlated and this relationship did not differ significantly between mitochondria located in synaptic or nonsynaptic regions of the axons from (H) (n=75, ANCOVA, F_1,72_=1.91, p=0.17). Significant post hoc differences are indicated as *p<0.05, **p<0.01, ***p<0.001, and ****p<0.0001.”